# USP37 prevents premature disassembly of stressed replisomes by TRAIP

Olga V. Kochenova[1,2,7], Giuseppina D'Alessandro [3,6,7], Domenic Pilger [4], Ernst Schmid [1], Sean L. Richards[3], Marcos Rios Garcia[5], Satpal S. Jhujh [5], Andrea Voigt[3], Vipul Gupta[3], Christopher J. Carnie [3], R. Alex Wu[1], Nadia Gueorguieva[3], Simon Lam [3], Grant S. Stewart [5], Johannes C. Walter [1,2] ✉ & Stephen P. Jackson [3] ✉

The eukaryotic replisome, which consists of the CDC45-MCM2-7-GINS (CMG) helicase, replicative polymerases, and several accessory factors, sometimes encounters proteinaceous obstacles that threaten genome integrity. These obstacles are targeted for removal or proteolysis by the E3 ubiquitin ligase TRAIP, which associates with the replisome. However, TRAIP must be carefully regulated to avoid inappropriate ubiquitylation and disassembly of the replisome. Here, we demonstrate that human cells lacking the de-ubiquitylating enzyme USP37 are hypersensitive to topoisomerase poisons and other replication stress-inducing agents. Furthermore, TRAIP loss rescues the hypersensitivity of *USP37* knockout cells to topoisomerase inhibitors. In *Xenopus* egg extracts depleted of USP37, TRAIP promotes premature CMG ubiquitylation and disassembly when converging replisomes stall. Finally, guided by AlphaFold-Multimer, we discovered that binding to CDC45 mediates USP37's response to topological stress. We propose that USP37 protects genome stability by preventing TRAIP-dependent CMG unloading when replication stress impedes timely termination.

Faithful DNA replication is essential to maintain genome integrity and prevent cancer and other diseases. Cells prepare for DNA replication in the G1 phase of the cell cycle, when ORC, CDC6, and CDT1 recruit MCM2-7 double hexamers to origins of DNA replication (licensing). In S-phase, cyclin-dependent kinases (CDKs) and DBF4-dependent kinases (DDKs) cooperate with additional factors to recruit the tetrameric GINS complex (SLD5, PSF1, PSF2, and PSF3) and CDC45 to MCM2-7, leading to the formation and activation of two CMG helicases. Active CMGs translocate in the 3' to 5' direction along the leading-strand DNA template. Once CMG unwinds the replication origin, replisomes are assembled, and bi-directional DNA synthesis commences. Replication finishes when replisomes from adjacent origins converge, leading to termination[1–3].

Replication termination is a highly regulated process that is critical for accurate genome maintenance[4]. To initiate termination, forks must merge, a process that depends critically on topoisomerase activity[5,6]. When converging CMGs meet, they pass each other, leading to the disengagement of the lagging strand template from the outer face of the CMG. Strand disengagement allows binding of the E3 ubiquitin ligase CRL2[Lrr1] to CMG, leading to MCM7 ubiquitylation and

[1]Department of Biological Chemistry and Molecular Pharmacology, Harvard Medical School, Blavatnik Institute, Boston, MA 02115, USA. [2]Howard Hughes Medical Institute, Boston, MA 02115, USA. [3]Cancer Research UK Cambridge Institute, Li Ka Shing Building, Robinson Way, Cambridge CB2 0RE, UK. [4]The Gurdon Institute and Department of Biochemistry, University of Cambridge, Cambridge, UK. [5]Institute of Cancer and Genomic Sciences, College of Medical and Dental Sciences, University of Birmingham, Birmingham, UK. [6]Present address: IFOM ETS, The AIRC Institute of Molecular Oncology, Milan, Italy. [7]These authors contributed equally: Olga V. Kochenova, Giuseppina D'Alessandro. ✉e-mail: johannes_walter@hms.harvard.edu; steve.jackson@cruk.cam.ac.uk

CMG extraction from chromatin by the p97 ATPase (aka VCP)[7–9]. In the presence of DNA damage, CMGs can also dissociate from chromatin. For example, if CMG encounters a nick in the leading-strand template, it slides off the end of the DNA, and if the nick is in the lagging strand template, lagging strand disengagement from CMG promotes its unloading by CRL2^Lrr1 (ref. 10). Thus, while CRL2^Lrr1 removes the bulk of CMGs from chromatin during a normal S-phase, it also removes CMG at specific types of DNA damage.

CMG unloading is also promoted by a second RING E3 ubiquitin ligase called TRAIP. TRAIP associates with replisomes and ubiquitylates CMGs that have converged on a DNA inter-strand cross-link (ICL), triggering their unloading by p97 and activation of ICL repair (Supplementary Fig. 1a-i and ref. 11). TRAIP also ubiquitylates DNA-protein cross-links (DPCs) that block replisome progression (Supplementary Fig. 1a-ii). and it might also act on RNAPII complexes during transcription-replication conflicts[12,13]. Together, these observations suggest that in S-phase, TRAIP ubiquitylates any proteinaceous structure encountered by the replisome ("*trans* ubiquitylation")[14]. At the same time, TRAIP appears to be unable to ubiquitylate the replisome with which it travels ("*cis* ubiquitylation"; Supplementary Fig. 1a-iii and ref. 12). This restriction of *cis* ubiquitylation is crucial to prevent premature CMG disassembly in S-phase, which would lead to fork collapse and incomplete DNA replication. Indeed, in most instances of replication stress, which are ultimately overcome, CMG is likely not unloaded so that replication can resume after the stress is resolved[15–18]. Notably, once cells enter mitosis, TRAIP supports *cis* ubiquitylation of CMGs, which disengages remaining replisomes to allow orderly chromosome segregation (Supplementary Fig. 1b; refs. 19–22). While TRAIP has the capacity for various forms of replisome-associated ubiquitylation, it remains unclear how aberrant TRAIP-mediated CMG ubiquitylation in interphase is avoided.

E3 ubiquitin ligases are counteracted by a family of ~100 deubiquitylases (DUBs)−specialized proteases that can cleave isopeptide bonds between ubiquitin and lysine residues, which removes or remodels ubiquitin chains[23]. Ubiquitin-specific proteases (USPs) comprise the largest subfamily of DUBs, and their aberrant expression has been linked to the occurrence and progression of cancer, rendering them promising targets for anticancer therapies[24,25]. In particular, the DUB USP37 has been linked to the DNA damage response (DDR), genome stability and DNA replication[26–29], and its overexpression provides a survival advantage to cancer cells[30–35]. Additional evidence points to USP37 as a possible replisome component[7,36,37], yet its specific role(s) at the replisome remain unclear.

Here, we report that USP37 protects cells from replication stress and that TRAIP loss rescues the hypersensitivity of *USP37* knockout cells to inhibitors of topoisomerases. Using *Xenopus* egg extracts, we demonstrate that TRAIP ubiquitylates CMGs when replication forks stall at DPCs and/or experience topological stress, and that USP37 counteracts TRAIP-mediated CMG-ubiquitylation. Furthermore, using in silico screening for protein–protein interactions and site-directed mutagenesis, we provide evidence that USP37 mediates its function by binding the CDC45 subunit of CMG. Based on these observations, we propose that USP37 promotes genome integrity and cell survival by preventing premature disassembly of the stressed replisome by TRAIP.

## Results
### CRISPR screening identifies USP37 as promoting camptothecin resistance
To identify novel regulators of DNA topological stress, we performed a genome-scale CRISPR-Cas9 gene-inactivation cell fitness screen[38] in human U2OS cells exposed to the TOP1 inhibitor camptothecin (Fig. 1a and Supplementary Fig. 2a). As expected, we identified factors involved in single-strand DNA break repair whose inactivation is known to cause camptothecin hypersensitivity, such as TDP1, PARP1, and XRCC1 (Fig. 1a). This screen also identified the DUB USP37. To extend

these results, we used CRISPR-Cas9 genome engineering to inactivate *USP37* in *TP53*-null human RPE-1 cells. Compared to USP37-proficient control cells (CTRL), two independent *USP37* knockout (KO) clones (10 and 19) that lacked detectable USP37 expression (Supplementary Fig. 2b) were hypersensitive to low doses of camptothecin (Fig. 1b). Additionally, these *USP37* knockout cells were hypersensitive to the TOP2 inhibitors etoposide and ICRF-193 (Fig. 1c, d), as well as the replicative DNA polymerase inhibitor aphidicolin (Fig. 1e). We further validated our observations in U2OS cells, where USP37 loss conferred a hypersensitivity towards camptothecin, etoposide, aphidicolin, and hydroxyurea (which impairs DNA replication by depleting dNTPs), but not the PARP1/2 inhibitor talazoparib that causes replication-associated DNA double-strand breaks (Supplementary Fig. 2c–h). Despite being classified as a common essential gene, *USP37* is not essential in U2OS or RPE-1 *TP53*-null cells, and no residual protein could be detected in the knockout clones (Supplementary Fig. 2b, c). Collectively, these results suggested that USP37 generally mitigates the deleterious effects of replication fork stalling, but not necessarily fork breakage.

To monitor the impact of USP37 loss on genome stability, we tested the levels of RPA and γH2AX foci, markers of ssDNA and DNA damage, respectively, in S-phase CTRL or *USP37* knockout cells (Fig. 1f–h). Upon treatment with camptothecin, *USP37* knockout cells showed increased RPA and γH2AX foci relative to CTRL cells. Similarly, treatment with aphidicolin, at doses that did not induce detectable damage in CTRL cells, strongly increased the number of RPA and γH2AX foci in *USP37* knockout cells, in line with published data[29]. Overall, these findings supported a role for USP37 in promoting genome integrity upon treatment with camptothecin or aphidicolin.

To address whether the catalytic activity of USP37 is necessary to modulate its cellular functions, we generated a catalytically inactive point mutant (USP37^C350A), as previously described[37], and expressed it in *USP37* knockout cells alongside the wild-type USP37 (USP37^WT) or an mCherry expressing vector (EV) (Supplementary Fig. 2i). Overexpression of USP37^WT restored the viability of the knockout cells upon treatment with camptothecin (Fig. 1i), etoposide (Fig. 1j), or aphidicolin (Fig. 1k), while overexpression of the inactive mutant did not. While these data indicated that the catalytic activity of USP37 protects cells from the deleterious effects of replication fork stalling, we cannot exclude the possibility that the lack of rescue by USP37^C350A may, in part, be due to an enhanced degradation of the mutant protein[39,40].

### Genetic screening unveils functional connections between USP37 and TRAIP
To identify factors that functionally interact with USP37, we performed parallel genome-scale CRISPR-Cas9 knockout screens in wild-type (WT) and *USP37* knockout cells and looked for genes whose loss affected fitness differentially in the two genetic backgrounds. TRAIP loss caused a fitness defect in WT but not *USP37* knockout U2OS cells (Fig. 2a). We validated this observation by performing cell-growth competition assays in Cas9-expressing WT and *USP37* knockout cells transiently transfected with a CRISPR single-guide RNA (sgRNA) targeting *TRAIP*. In line with previous reports[13,41,42], *TRAIP* inactivation was not well tolerated in WT cells, as indicated by a marked reduction over time of the cell population harboring mutations at the targeted *TRAIP* locus ("edited"). By contrast, such a reduction in the *TRAIP* edited population was not evident in the *USP37* knockout cells (Fig. 2b and Supplementary Fig. 3a). This epistatic relationship between *TRAIP* and *USP37* suggested that TRAIP and USP37 control a common pathway.

Considering the above data, we tested whether TRAIP loss might affect the hypersensitivity of *USP37* knockout cells to camptothecin. To this end, we generated polyclonal knockouts of *TRAIP* in WT and *USP37* knockout *TP53*-null human RPE-1 cells (Supplementary Fig. 3b) and subjected them to clonogenic survival assays. While *TRAIP* knockout and *USP37* knockout cells were hypersensitive to camptothecin, as expected[27,43], the *USP37-TRAIP* double knockout cells largely lost their

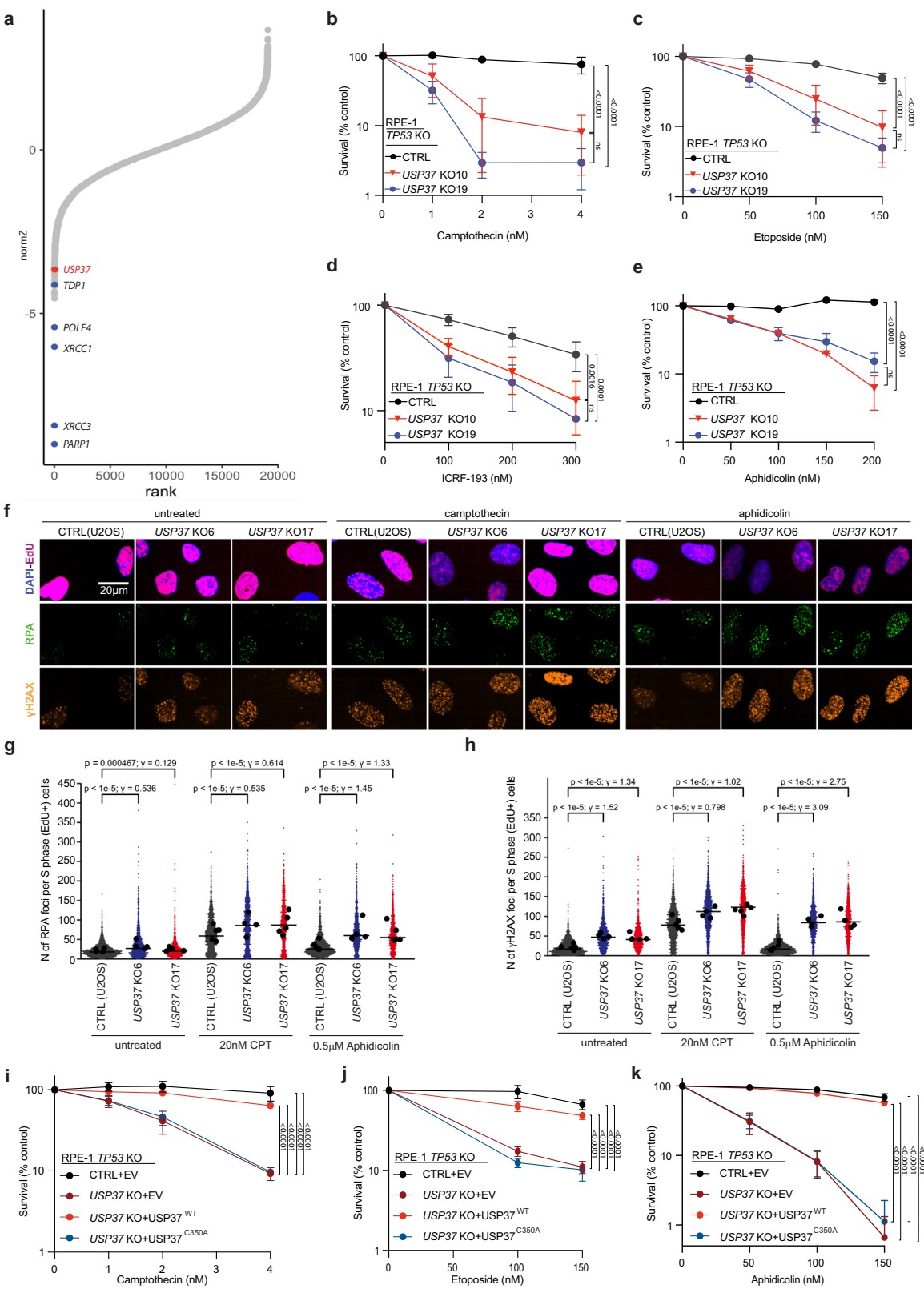

hypersensitivity towards camptothecin (Fig. 2c and Supplementary Fig. 4a), indicating that TRAIP loss compensates for the absence of USP37 and vice versa. Furthermore, while loss of USP37 or TRAIP alone caused increased sensitivity to ICRF-193, cells lacking both factors were more resistant than *USP37* knockout cells, and equally sensitive to the *TRAIP* knockout alone (Fig. 2d and Supplementary Fig. 4b).

These findings indicated that the main source of toxicity in *USP37* knockout cells upon treatment with camptothecin or ICRF-193 is due to TRAIP. In contrast, *TRAIP* loss did not restore *USP37* knockout cells' tolerance of aphidicolin (Fig. 2e and Supplementary Fig. 4c), suggesting a different cause of toxicity when USP37 is lost in this setting. TRAIP depletion significantly reduced the number of camptothecin-induced

**Fig. 1 | USP37 loss protects cells from topoisomerase poisons- and replication stress-associated DNA damage. a** Rank-plot showing gene enrichment scores (normZ) of CRISPR screen hits upon camptothecin treatment in human U2OS cells. Negative scores represent dropouts, genes whose loss is predicted to promote drug sensitivity. Blue dots indicate well-characterized factors affecting cellular sensitivity to camptothecin; the red dot indicates *USP37*. **b–e** Clonogenic survival assays of control (CTRL) and *USP37* knockout (KO; results from two independent clones plotted) RPE-1 *TP53* KO cells upon treatment with (**b**) camptothecin, (**c**) etoposide, (**d**) ICRF-193, or (**e**) aphidicolin. *n* = 4 independent experiments (**b–d**), *n* = 3 independent experiments (**e**). Statistical analysis for (**b–e**) was performed using two-way ANOVA and Tukey test for multiple comparisons. **f** Representative images of RPA (green) and γH2AX (orange) staining in S-phase (magenta) positive cells. Scale bar: 20 µm. **g, h** Quantification of RPA (**g**) or γH2AX (**h**) foci in S-phase cells (EdU positive). Cells were treated with the indicated doses of the drugs for 4 h. For (**g**), untreated: *n* = 2427 (WT), 1599 (*USP37* KO6), 1212

(*USP37* KO17) cells; camptothecin: *n* = 2316 (WT), 1665 (*USP37* KO6), 1350 (*USP37* KO17) cells; aphidicolin: *n* = 2538 (WT), 1234 (*USP37* KO6), 1164 (*USP37* KO17) cells. For (**h**), same as for (**g**), but *n* = 2426 (WT, untreated), 1598 (*USP37* KO6, untreated) cells. Cells were analyzed from five biological replicates, except for *USP37* KO6 (four biological replicates). Bars represent the median. Black dots indicate means in each independent experiment. Statistics indicate two-tailed Wilcoxon rank-sum tests for *p* values without adjustment for multiple hypothesis testing and Cohen's d for effect sizes (γ) (see methods). CPT camptothecin. **i–k** Clonogenic survival assays of control (CTRL) or *USP37* KO (clone 10) cells expressing mCherry (EV), mCherry-USP37$^{WT}$ or mCherry-USP37$^{C350A}$ (catalytic inactive) upon treatment with camptothecin (**i**), etoposide (**j**), or aphidicolin (**k**); *n* = 4 independent experiments (**i**); *n* = 5 independent experiments (**j**), with the exception of "C350A" condition, where *n* = 4); *n* = 3 independent experiments (**k**). Bars represent means ± SEM. Statistical analysis for (**i–k**) was performed using two-way ANOVA and Šídák test for multiple comparisons. Source data are provided as a Source Data file.

RPA and γH2AX foci in CTRL cells (Fig. 2f–h and Supplementary Fig. 4d), consistent with previous findings[44]. This reduction was even more pronounced in *USP37* knockout cells, bringing the levels of RPA and γH2AX foci down to those observed in WT cells. Furthermore, TRAIP depletion fully restored the fork progression defect observed in *USP37* knockout cells upon treatment with camptothecin (Supplementary Fig. 4e). Overall, these results suggested that USP37 is required for cells to overcome diverse forms of replication stress, and that in the context of topological stress caused by camptothecin or ICRF-193, its primary function is to counteract the action of TRAIP.

## USP37 prevents CMG disassembly by TRAIP upon TOP2α inhibition

Prior work showed that TRAIP ubiquitylates CMG[11,20–22]. We therefore explored whether USP37 counteracts TRAIP-dependent CMG ubiquitylation under conditions of topological stress. To this end, we replicated circular plasmids in *Xenopus laevis* egg extracts (Supplementary Fig. 5a), which recapitulate TRAIP- and CRL2$^{Lrr1}$-dependent CMG ubiquitylation[7,11]. We first confirmed that in this system the TOP2 inhibitor ICRF-193 greatly delayed CMG unloading (Fig. 3a, compare lanes 1–3 and 7–9; note different time courses in the two conditions), as expected from the fact that it impairs replisome convergence (Supplementary Fig. 5b; ref. 6). Contrasting with ICRF-193 treatment, TOP2α depletion (Supplementary Fig. 5c) had only a mild effect on CMG unloading (Supplementary Fig. 5d, lanes 1–3 vs. 7–9), probably because fork rotation or accessory helicases allowed fork convergence in this setting[5,6,45,46]. Moreover, TOP2α immuno-depletion abrogated the strong effect of ICRF-193 on CMG unloading (Supplementary Fig. 5d, lanes 4–6 vs. 10–12). Together with the fact that TOP2α can accumulate ahead of the replisome (Supplementary Fig. 5e), these data strongly suggested that ICRF-193 delays fork convergence and CMG unloading primarily by trapping TOP2α complexes ahead of the replication fork.

To test whether USP37 affects CMG unloading in these settings, we immunodepleted USP37 from egg extracts (Supplementary Fig. 6a). While USP37 depletion sometimes modestly enhanced CMG unloading in unperturbed reactions (Supplementary Fig. 6b, lanes 1–6), its absence consistently accelerated CMG disassembly in the presence of ICRF-193 (Fig. 3a–c and Supplementary Fig. 6b, d, f, i). By contrast, USP37 depletion did not accelerate CMG unloading in extracts depleted of TOP2α, whether or not ICRF-193 was added (Supplementary Fig. 5d, lanes 7–12 vs. 13–18). Premature CMG unloading was reduced by supplementing USP37-depleted extracts with WT HA-tagged USP37 (USP37$^{WT}$; expressed in a transcription-translation extract), but not with catalytically inactive USP37 (USP37$^{C347S}$; Fig. 3b, lanes 2, 4, 6, 8 and Supplementary Fig. 6c). In the presence of the p97 inhibitor NMS-873 (p97-i), USP37 depletion stimulated polyubiquitylation of MCM7 and MCM4 on chromatin, which was most evident from loss of the unmodified forms of these proteins

(Fig. 3b, lane 10 vs. 12). Furthermore, this effect was reversed by USP37$^{WT}$ but not USP37$^{C347S}$ (Fig. 3b, lanes 14 and 16). These results argued that when replisomes encounter trapped TOP2α complexes, USP37 suppresses CMG ubiquitylation and p97-dependent unloading.

We next tested whether CMG unloading in the absence of USP37 depends on TRAIP. Consistent with this idea, depletion of TRAIP (Supplementary Fig. 6e) inhibited premature CMG unloading in USP37-depleted extracts containing ICRF-193 (Fig. 3c, lanes 4 vs. 6). Moreover, compared to controls, TRAIP depletion increased the proportion of unmodified MCM7 (Fig. 3c, lane 12 vs. 14), with re-addition of TRAIP to such extracts rescuing CMG unloading and ubiquitylation (Fig. 3c, lanes 8 and 16). We speculate that the remaining TRAIP-independent ubiquitylation of MCM7 (Fig. 3c, lane 14) is CRL2$^{Lrr1}$-dependent and occurred due to termination of some forks despite the presence of ICRF-193 (ref. 6). USP37 depletion also induced TRAIP-dependent CMG ubiquitylation and unloading when we replicated sperm chromatin instead of plasmid DNA in the presence of ICRF-193 (Supplementary Fig. 6h–k). We conclude that inhibition of TOP2α in the absence of USP37 activity induces premature unloading of CMG on diverse replication substrates.

To address whether USP37 controls CMG unloading in mammalian cells, we employed the SIRF (in situ Protein Interaction with Nascent DNA Replication Forks) assay[47], which uses a fluorescent-based proximity ligation assay (PLA) to probe the proximity between a protein of interest and EdU-labeled replicating DNA. A PLA signal is only visible when an antibody against the protein of interest and an antibody against biotin, which detects biotinylated EdU, are located within 40 nm of each other. When using the number of PLA foci as a readout of CDC45 localization at sites of DNA replication, we observed that while the EdU signal was similar among all the conditions tested (Supplementary Fig. 7a), camptothecin strongly reduced the number of PLA foci in control cells (Fig. 3d, conditions I vs. VII) and even further in *USP37* knockout cells (conditions VII vs. IX and XI). Notably, TRAIP loss prevented the decline in CDC45 PLA signal in control and in *USP37* knockout cells (conditions VIII, X, XII). We next monitored the ability of replication forks to restart following the removal of camptothecin. While both *TRAIP* knockout and *USP37* knockout cells had elevated levels of stalled forks that failed to restart, *USP37-TRAIP* knockout reduced fork stalling to the level seen in WT cells after camptothecin washout (Supplementary Fig. 7b). Collectively, these results indicated that when forks experience stress in egg extracts and in mammalian cells, USP37 prevents CMG disassembly by TRAIP.

## USP37 suppresses TRAIP-dependent disassembly of converged CMGs

When TOP2α is inhibited in *Xenopus* egg extracts, converging replisomes stall several hundred base pairs (bp) apart, followed by slow fork merging[6]. We therefore hypothesized that in the presence of ICRF-193, TRAIP ubiquitylates CMGs that stall close to each other. To test

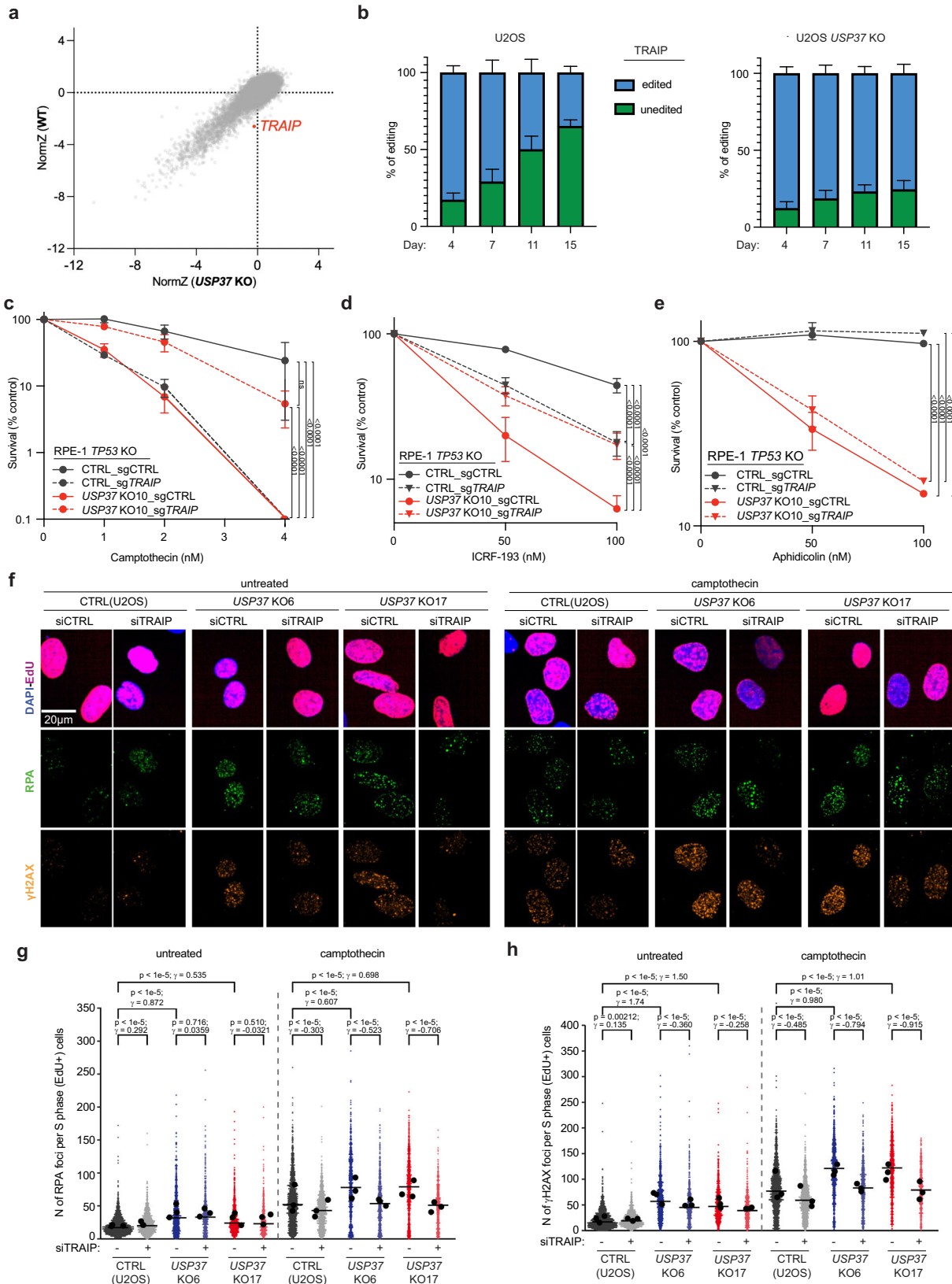

this idea, we created a DNA substrate in which two tandem methylated HpaII DNA-protein cross-links (meDPCs) on the leading strands were placed 165 bp apart (Fig. 4a, top). Such tandem meDPCs cannot be degraded or bypassed by CMG and therefore pose a strong barrier to the replisome, reducing CMG unloading (ref. 48; Fig. 4a, lanes 1–2). However, when USP37 was depleted, the CMGs stalled at these tandem

DPCs underwent unloading (Fig. 4a, compare lanes 2 and 4) and polyubiquitylation in the presence of p97-i (Fig. 4a, lanes 8 and 10). This CMG unloading and ubiquitylation was attenuated when TRAIP was also absent (Fig. 4a, lanes 6 and 12). These data thus indicated that USP37 suppresses replisome ubiquitylation and disassembly by TRAIP when CMGs stall at close range for an extended time.

**Fig. 2 | Genetic screen unveils functional connections between USP37 and TRAIP. a** Biplot showing gene enrichment scores (normZ) reflecting viability in untreated conditions of WT (y-axis) and *USP37* KO (x-axis) U2OS cells. **b** U2OS *USP37* WT cells (left histogram) and *USP37* KO cells (right histogram) were transfected with a CRISPR sgRNA targeting *TRAIP* and samples were collected 4, 7, 11, and 15 days afterwards and subjected to *TRAIP* sequence analysis. Percentages of unedited or edited cells at the *TRAIP* locus at the indicated time points are plotted. $n = 3$ independent experiments. Bars represent means ± SEM. **c–e** Clonogenic survival assays of CTRL or *USP37* KO RPE-1 *TP53* KO cells transduced with a LacZ control sgRNA or with a sgRNA targeting *TRAIP* upon treatment with camptothecin (**c**), ICRF-193 (**d**), and aphidicolin (**e**). $n = 3$ independent experiments. Bars represent means ± SEM. Half plot points indicate zero percent viability. Statistical analysis for (**c–e**) was performed using two-way ANOVA and Tukey test for multiple

comparisons. **f** Representative images of RPA (green) and γH2AX (orange) staining in S-phase (magenta) positive cells. Scale bar: 20 μm. **g, h** Quantification of RPA (**g**) or γH2AX (**h**) foci in S-phase cells. Cells were treated with 20 nM camptothecin for 4 h. For (**g, h**), untreated: $n = 1747$ (CTRL, siCTRL), 1262 (CTRL, siTRAIP), 865 (*USP37* KO6, siCTRL), 533 (*USP37* KO6, siTRAIP), 753 (*USP37* KO17, siCTRL), 458 (*USP37* KO17, siTRAIP); camptothecin: $n = 1885$ (CTRL, siCTRL), 1315 (CTRL, siTRAIP), 838 (*USP37* KO6, siCTRL), 622 (*USP37* KO6, siTRAIP), 806 (*USP37* KO17, siCTRL), 476 (*USP37* KO17, siTRAIP). Cells were analyzed from three biological replicates. Bars represent median. Black dots indicate means in each independent experiment. Statistics indicate two-tailed Wilcoxon rank-sum tests for $p$ values without adjustment for multiple hypothesis testing and Cohen's d for effect sizes (γ) (see methods). Source data are provided as a Source Data file.

To explore the distance-dependence of CMG ubiquitylation, we generated a panel of meDPC substrates in which DPCs were separated by 56, 305, and 1033 bp (Supplementary Fig. 8c, top). All tested substrates led to stalling of converging CMGs in mock-depleted extracts, as seen from the persistence of CMG on chromatin at the late time point (Supplementary Fig. 8c, lanes 2, 6, 10). Strikingly, USP37 depletion enhanced CMG disassembly even when meDPCs were spaced ~1000 bp apart (Supplementary Fig. 8c, lanes 10 vs. 12). Likewise, the ubiquitylation of MCM7 was similarly enhanced upon USP37 depletion in the context of all substrates (Supplementary Fig. 8c, lanes 13-24). Furthermore, in the context of USP37 depletion, the co-depletion of TRAIP blocked CMG ubiquitylation and unloading even when CMGs were 1 kb apart (Fig. 4b, lanes 3–4 vs. 5–6, and 11–12 vs. 13–14), and these effects were rescued by re-addition of TRAIP (Fig. 4b, lanes 8 and 16). Collectively, these results showed that in the absence of USP37, TRAIP can induce unloading of CMGs even when they are stalled at a considerable distance from each other.

### USP37 counteracts *trans* ubiquitylation of CMG by TRAIP

We showed previously that in interphase extracts, TRAIP associates with the replisome and ubiquitylates proteins ahead of the replication fork "in trans" while being unable to ubiquitylate the CMG that it travels with "*in cis*" (Supplementary Fig. 1a; refs. 11,12). It was therefore unexpected that in USP37-depleted interphase extracts, TRAIP was able to ubiquitylate CMGs that are stalled ~1 kb apart. This observation raised the possibility that in the absence of USP37, TRAIP acquires the capacity to promote *cis* ubiquitylation of CMG, as seen in mitotic extracts (Supplementary Fig. 1b). To test this idea, we investigated whether, in the absence of USP37, TRAIP can ubiquitylate *in cis* terminated CMGs that are normally only ubiquitylated by CRL2^LrrI when CMGs have passed each other after fork convergence[11,21]. Thus, we replicated DNA in the presence of p97-i to prevent CMG unloading during termination, and we also added the general Cullin inhibitor MLN-4924 (Cul-i) to minimize CRL2^LrrI-dependent ubiquitylation of CMG (Fig. 4c). As expected, Cul-i greatly reduced MCM7 ubiquitylation in mock-depleted extracts (Fig. 4c, lanes 1 and 4), but residual ubiquitylation was still observed, possibly due to incomplete inhibition of CRL2^LrrI (Fig. 4c, lane 4) and/or the action of another ubiquitin E3 ligase. Although USP37 depletion enhanced this residual ubiquitylation of MCM7 (Fig. 4c, lane 5), most MCM7 remained unmodified, and importantly, the residual ubiquitylation was not dependent on TRAIP (Fig. 4c, lane 6). These data suggested that although TRAIP associates with terminated CMGs[7], it is unable to support *cis* ubiquitylation, even in the absence of USP37. This result further implied that in the absence of USP37, TRAIP promotes ubiquitylation of distant CMGs in trans, perhaps as a result of replisome co-localization[49].

To further explore the mechanism of how stressed replisomes are ubiquitylated in the absence of USP37, we employed a DNA substrate containing an array of 32 *lac* operator (*lacO*) repeats (comprising 990 bp) to which we bound the *lac* repressor (LacR). LacR-arrays slow CMG progression by stalling the replisome, and this

effect is enhanced when the accessory helicase RTEL1 is depleted (refs. 21,50, Supplementary Fig. 9c and refs. 12,48). When the LacR-bound plasmid DNA was replicated in egg extract immunodepleted of both USP37 and RTEL1, stalled CMGs were not unloaded (Fig. 4d, lanes 5–8), and they underwent only a very low level of ubiquitylation (Fig. 4e, lanes 5–8). This contrasted with results obtained with the DPC-containing substrate in the same extract (Fig. 4d, e, lanes 1–4). These results imply that local chromatin structure influences whether CMGs are ubiquitylated in the absence of USP37, as might be expected for a *trans* ubiquitylation mechanism that relies on replisome co-localization. Importantly, in mitotic extracts, CMGs stalled at DPCs or the LacR array were unloaded and hyper-ubiquitylated with similar efficiencies independently of USP37 (Fig. 4d, e, lanes 9–16; Supplementary Fig. 9j). Collectively, our results suggest that TRAIP-mediated CMG ubiquitylation at long range in interphase involves a *trans* ubiquitylation mechanism that depends on higher order chromatin structure, possibly to allow co-localization of replisomes.

### Evidence that USP37 functions by binding CDC45

Previous experiments suggested that USP37 is a component of the replisome, and we wanted to understand how USP37 is recruited to this molecular machine[7,36,37]. However, we did not detect USP37 on replicating DNA in plasmid pull-down experiments in egg extracts (Supplementary Fig. 10a). We speculated this is because plasmid pull-down is a lengthy procedure during which loosely bound proteins dissociate from chromatin[51]. To circumvent this issue, we used a rapid sperm chromatin isolation assay[52]. In this setting, we detected USP37 on chromatin, as seen previously[7], and this binding was abolished by the replication-licensing inhibitor geminin and by inhibitors of CDK and DDK kinases, which are required to convert pre-replication complexes into active CMG helicases (Fig. 5a). We thus concluded that USP37 loads onto chromatin in a manner dependent on CMG assembly.

We next explored how USP37 associates with replisomes by using AlphaFold-Multimer (AF-M) to screen in silico for potential USP37 binding partners among 285 proteins involved in genome maintenance (predictomes.org; ref. 53). We scored the results using a classifier called SPOC (structure prediction and omics classifier) that is trained to distinguish between high and low confidence AF-M predictions[53]. According to this metric, cohesin subunits (STAG2, STAG1, and SMC3) as well as replication factors, RPA2 and CDC45, were the top-ranked proteins predicted to bind USP37 (Supplementary Data 1 and see ref. 53). Since USP37 interaction with cohesin was reported previously[37], we tested whether, like USP37, cohesin suppresses premature CMG unloading. However, depletion of SMC3 (Supplementary Fig. 10c) had no measurable effect on CMG unloading at meDPCs (Supplementary Fig. 10d, compare lanes 2,4, and 6), arguing against a role for cohesin in recruiting USP37 to the replisome. Since USP37 still associates with terminated replisomes in the absence of ssDNA[7], RPA2 was also unlikely to recruit USP37.

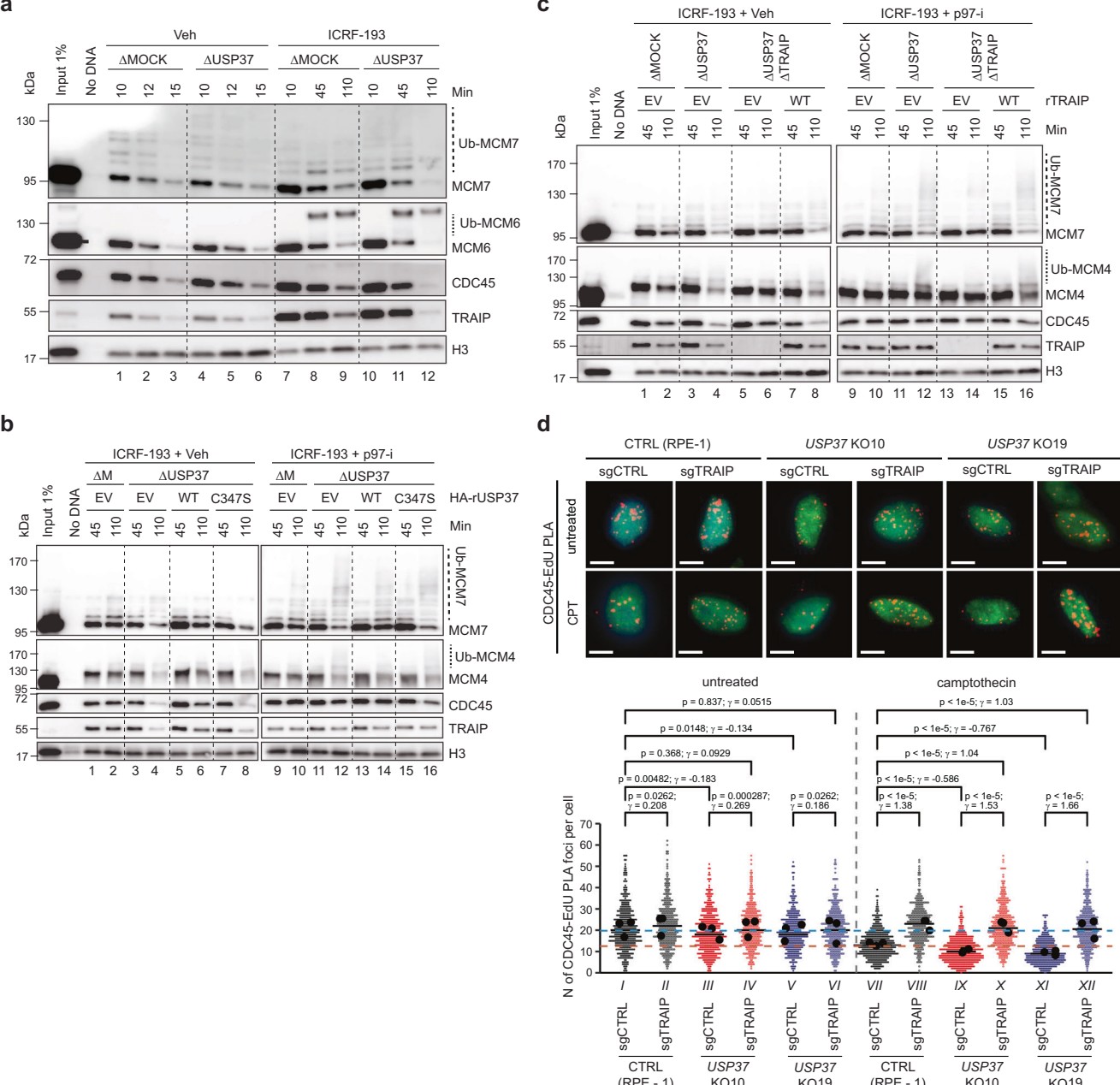

**Fig. 3 | USP37 depletion induces premature CMG unloading by TRAIP during topological stress. a** Plasmid DNA was incubated in the indicated egg extracts in the presence or absence of ICRF-193. At specified times, chromatin was recovered and immunoblotted for the indicated proteins. USP37 depletion efficiency is shown in Supplementary Fig. 6a. Ub-MCM7, ubiquitylated MCM7; Ub-MCM6, ubiquitylated MCM6; Ub-CDC45, ubiquitylated CDC45. See also Supplementary Fig. 6b. **b** Plasmid DNA was incubated in the indicated egg extracts in the presence of ICRF-193 and, where indicated, p97-i. Extracts were supplemented with recombinant WT or catalytically inactive (C347S) USP37 expressed in wheat germ extract (WGE), or WGE with empty vector (EV). At specified times, chromatin was recovered and immunoblotted for the indicated proteins. Samples are from the same experiment; blots were processed in parallel. ΔM, ΔMOCK; Ub-MCM4, ubiquitylated MCM4. See also Supplementary Fig. 6c, d. **c** As in (**b**), but where indicated, TRAIP was co-depleted with USP37. Egg extracts were supplemented with recombinant WT TRAIP expressed in WGE, or WGE with EV. Samples are from the same experiment; blots were processed in parallel. Depletion efficiencies and levels of recombinant

proteins are shown in Supplementary Fig. 6e. See also Supplementary Fig. 6f. **d** Representative pictures of CDC45-EdU PLA. Scale bar: 10 μm (top); Dot plot indicating the number of PLA foci between CDC45 and EdU (replicating DNA) in untreated or camptothecin-treated cells (bottom). Untreated: *n* = 441 (CTRL, sgCTRL), 389 (CTRL, sgTRAIP), 457 (*USP37* KO10, sgCTRL), 363 (*USP37* KO10, sgTRAIP), 386 (*USP37* KO19, sgCTRL), 385 (*USP37* KO19, sgTRAIP) cells; camptothecin: *n* = 450 (CTRL, sgCTRL), 361 (CTRL, sgTRAIP), 449 (*USP37* KO10, sgCTRL), 409 (*USP37* KO10, sgTRAIP), 369 (*USP37* KO19, sgCTRL), 372 (*USP37* KO19, sgTRAIP) cells. Cells were analyzed from three biological replicates. Bar represents the median of three independent experiments. Dashed cyan and brown lines are the position of the median for untreated and camptothecin-treated CTRL, sgCTRL cells, respectively. Black dots indicate means in each independent experiment. Statistics indicate two-tailed Wilcoxon rank-sum tests for *p* values without adjustment for multiple hypothesis testing and Cohen's d for effect sizes (γ) (see methods). Source data are provided as a Source Data file.

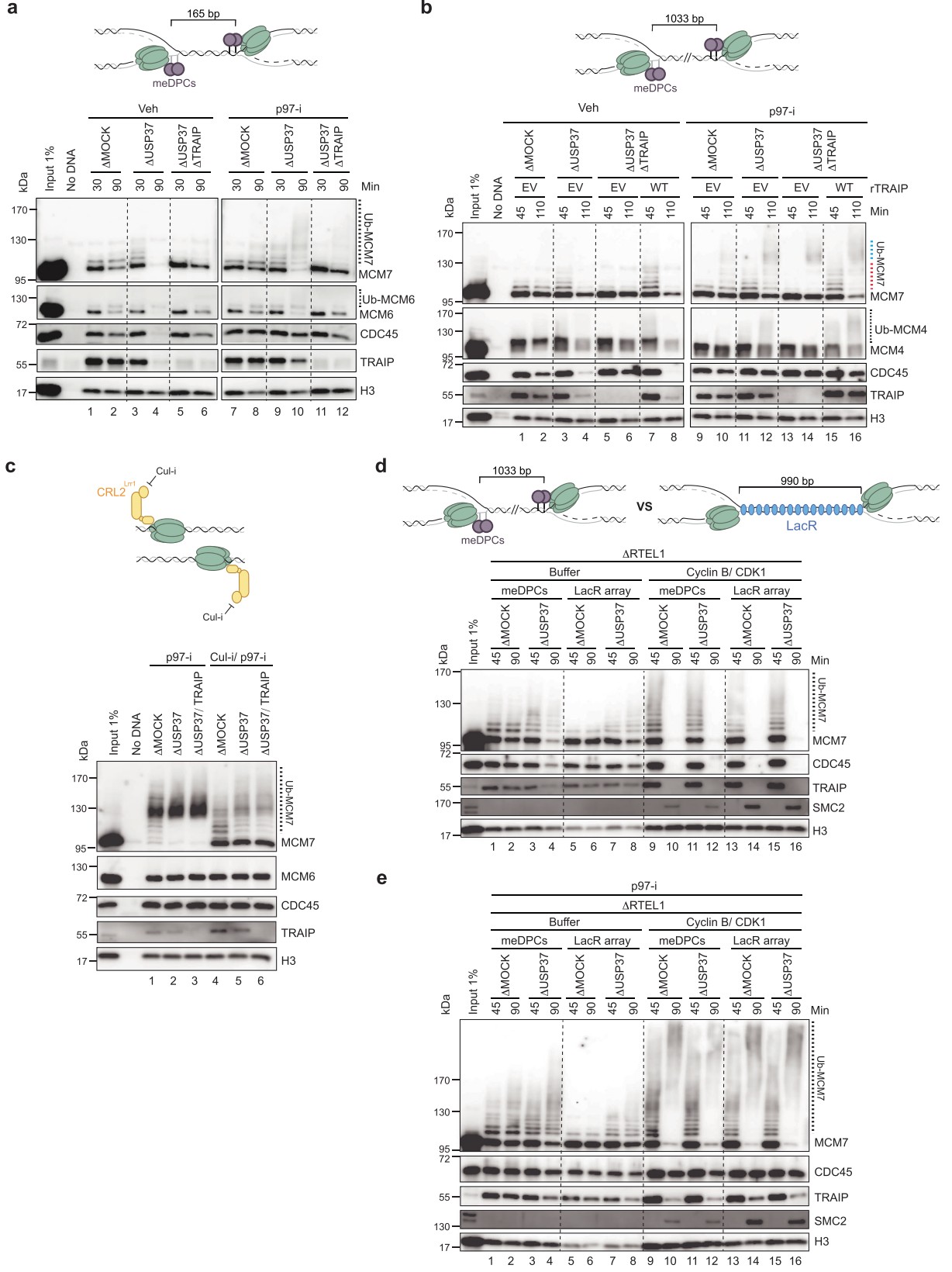

We therefore focused on the core replisome component, CDC45, which ranked fourth among potential interactors. Conversely, USP37 was ranked fourth among 285 genome maintenance proteins predicted to bind CDC45 (Supplementary Data 1 and Supplementary Fig. 10e, f). Similar results were obtained when *Xenopus* USP37 was folded in silico with *Xenopus* replication proteins

(Supplementary Data 1). AF-M predicted that an antiparallel β-sheet of the USP37 N-terminal pleckstrin homology (PH) domain binds CDC45 (Supplementary Fig. 10e–g) adjacent to DNA polymerase epsilon subunits DPOE1 and DPOE2 without clashing with any known replisome proteins (Fig. 5b). Furthermore, consistent with the USP37-CDC45 interaction having an important function, the predicted

**Fig. 4 | USP37 depletion induces TRAIP-dependent disassembly of CMGs converged at tandem DPCs, but not at terminated forks or ones separated by a LacR array. a** Top, a schematic of the DNA-protein cross-link substrate. meDPCs, methylated DNA-protein cross-links. The square bracket indicates the distance between distal leading-strand DPCs. Bottom, the meDPCs substrate was incubated in the indicated egg extracts in the presence or absence of p97-i. At specified times, chromatin was recovered and immunoblotted for the indicated proteins. Samples are from the same experiment; blots were processed in parallel. See also Supplementary Fig. 8a, b. **b** Same as (**a**), top, but the distance between distal meDPCs was 1033 bp. Bottom, the meDPCs substrate was incubated in the indicated egg extracts in the presence or absence of p97-i. Extracts were supplemented with recombinant WT TRAIP expressed in WGE or WGE with empty vector (EV). Samples are from the same experiment; blots were processed in parallel. Dashed blue line, probable

CRL2$^{Lrr1}$-dependent ubiquitylation resulting from termination events. Dashed red line, TRAIP-dependent ubiquitylation. See also Supplementary Fig. 8e, f. **c** Top, schematic showing terminated CMGs and inhibited CRL2$^{Lrr1}$ by Cul-i. Bottom, plasmid DNA was replicated in the indicated egg extracts with p97-i and, where indicated, with Cul-i. At 90 min, chromatin was recovered and immunoblotted for the indicated proteins. See also Supplementary Fig. 9a, b. **d** Top, a schematic of CMGs converged at the meDPCs versus the LacR-bound *lacO* array. Bottom, plasmid DNA containing a 32x *lacO* array was preincubated with LacR and then replicated in the indicated egg extracts. At 30 min, reactions were supplemented with buffer or Cyclin B/CDK1. At the specified times, chromatin was recovered and immunoblotted for the indicated proteins. See also Supplementary Fig. 9e, f. **e** Same as (**d**), but at 40 min, reactions were supplemented with p97-i. See also Supplementary Fig. 9g, i. Source data are provided as a Source Data file.

interacting residues within the left and right binding interfaces of USP37 are highly conserved between human and *Xenopus* (Supplementary Fig. 10h–j). Because the USP37 catalytic domain is connected to the PH domain via an ~800 Å unstructured region, we infer that USP37 should be able to reach and de-ubiquitylate most components of the replisome (Fig. 5b).

To test the function of the predicted USP37-CDC45 interaction, we made mutations in four highly conserved USP37 residues in the predicted left (R11A, N13A, K21A, and W22A) or right interfaces (K63A, R71A, D85A, and K86A), or in all eight residues, generating USP37$^{4A-L}$, USP37$^{4A-R}$, and USP37$^{8A}$, respectively (Supplementary Fig. 10j). These amino acid changes caused a minor reduction of the catalytic activity of USP37, as monitored by cleavage of K48-linked tetraubiquitin (Supplementary Fig. 10k, l). Unlike the WT protein, USP37$^{4A-L}$, USP37$^{4A-R}$, and USP37$^{8A}$ did not bind chromatin efficiently (Fig. 5c, lane 7–10), and they failed to prevent premature CMG ubiquitylation and unloading in the presence of ICRF-193 in egg extract (Fig. 5d). Furthermore, in mammalian cells, USP37$^{WT}$ but not USP37$^{8A}$ co-precipitated CDC45 and other CMG components (Fig. 5e).

To further assess USP37 localization to the replisome in mammalian cells, we employed SIRF, which detected a basal level of USP37 at sites of DNA replication in untreated conditions. Upon treatment with camptothecin, USP37$^{WT}$ and catalytically inactive USP37$^{C350}$ localization at replicating DNA increased, while USP37$^{8A}$ did so to a lesser extent (Fig. 5f and Supplementary Fig. 12a). Furthermore, we found that USP37$^{WT}$ but not USP37$^{8A}$ complemented the camptothecin and etoposide hypersensitivities of *USP37* knockout cells (Fig. 5g, h and Supplementary Figs. 2i, 12b, c), and the sensitivity of USP37$^{8A}$ cells to camptothecin was suppressed by loss of TRAIP (Fig. 5i and Supplementary Fig. 12d). Finally, USP37$^{WT}$ but not USP37$^{C350}$ or USP37$^{8A}$ increased CDC45 localization to sites of DNA replication and replication fork progression upon camptothecin treatment (Supplementary Fig. 12e, f). Collectively, the data suggested that USP37's interaction with CDC45 counteracts TRAIP-mediated CMG ubiquitylation, which mitigates the toxic effects of topoisomerase inhibitors.

## Discussion

Replisomes are actively disassembled during replication termination and at ICLs, yet how cells prevent inadvertent loss of CMG before termination and at other obstacles is incompletely understood. Based on experiments in cell-free extracts and mammalian cells, we report here that under certain forms of replication stress, USP37 prevents CMG ubiquitylation and unloading by the TRAIP E3 ubiquitin ligase. We propose that this regulation is critical to allow the completion of DNA replication and the suppression of genome instability.

We uncovered a complex genetic interaction between *USP37* and *TRAIP*. While *USP37* knockout cells were hypersensitive to TOP1 and TOP2 inhibitors, this phenotype was ameliorated in *TRAIP/USP37* double knockouts, indicative of genetic suppression or "synthetic rescue". Consistent with the cell viability data, we observed that the increased number of camptothecin-induced RPA

and γH2AX foci in *USP37* knockout cells was reversed by TRAIP loss, indicating that USP37 protects against the accumulation of ssDNA and DSBs resulting from aberrant TRAIP activity. The simplest interpretation of these results is that in the absence of USP37, hyperubiquitylation of one or more TRAIP substrates leads to DNA damage and cell death.

Conversely, the camptothecin sensitivity in *TRAIP* knockouts was largely suppressed by ablating *USP37*, demonstrating synthetic rescue in the other direction. Similarly, while our genetic screens revealed that TRAIP loss reduced the fitness of WT cells in the absence of external stress, as observed previously[13,42], in USP37-deficient cells, the loss of TRAIP had no further deleterious effect. These results suggest that TRAIP has important targets whose ubiquitylation is critical for cell viability in the face of endogenous and exogenous stress. We speculate that when USP37 is lost in the context of TRAIP deficiency, a different E3 ubiquitin ligase whose activity is normally restrained by USP37, can take over TRAIP's function. Thus, the interplay of TRAIP and USP37 is complex, and we focus here on the toxic activity of TRAIP in the absence of USP37. Notably, TRAIP mutations are associated with microcephalic dwarfism, a condition characterized by diminished growth and reduced brain size[54]. We speculate that inactivating USP37 or inhibiting its activity might ameliorate phenotypes associated with this condition.

Our observations in *Xenopus* egg extracts help explain the genetic interaction between TRAIP and USP37. In this cell-free system, resolution of topological stress is essential to allow rapid replisome convergence and unloading[6]. Notably, we found that when replisome convergence was delayed by the TOP2 inhibitor ICRF-193, USP37 suppressed TRAIP-dependent premature CMG unloading. Similarly, when we stalled forks with DNA-protein cross-links, USP37 prevented TRAIP-dependent CMG unloading. Based on these observations, we propose that when converging forks stall, USP37 prevents CMG unloading by TRAIP (Fig. 6), and we speculate that a similar phenomenon occurs when mammalian cells are treated with topoisomerase inhibitors.

We also found that TRAIP promotes CMG unloading in USP37-depleted extracts even when forks were stalled ~1 kb apart. This observation was surprising because we previously showed that in interphase, TRAIP normally only ubiquitylates proteins ahead of the fork (*trans* ubiquitylation) while being unable to ubiquitylate the CMG with which it travels (*cis* ubiquitylation) (Supplementary Fig. 1). How can TRAIP ubiquitylate CMGs separated by 1 kb in the absence of USP37? While *cis* ubiquitylation occurs in mitosis, we do not believe that USP37 suppresses *cis* ubiquitylation in interphase because even in USP37-depleted extracts, TRAIP was not able to ubiquitylate terminated replisomes that passed each other. Instead, we hypothesize that when converging forks stall at distant DPCs, positive supercoiling, chromatin compaction, and/or physical coupling of replisomes bring the CMGs close enough together to allow *trans* ubiquitylation by TRAIP. Indeed, the spatial and functional interaction of two converging replisomes has been proposed to occur constantly in S-phase in

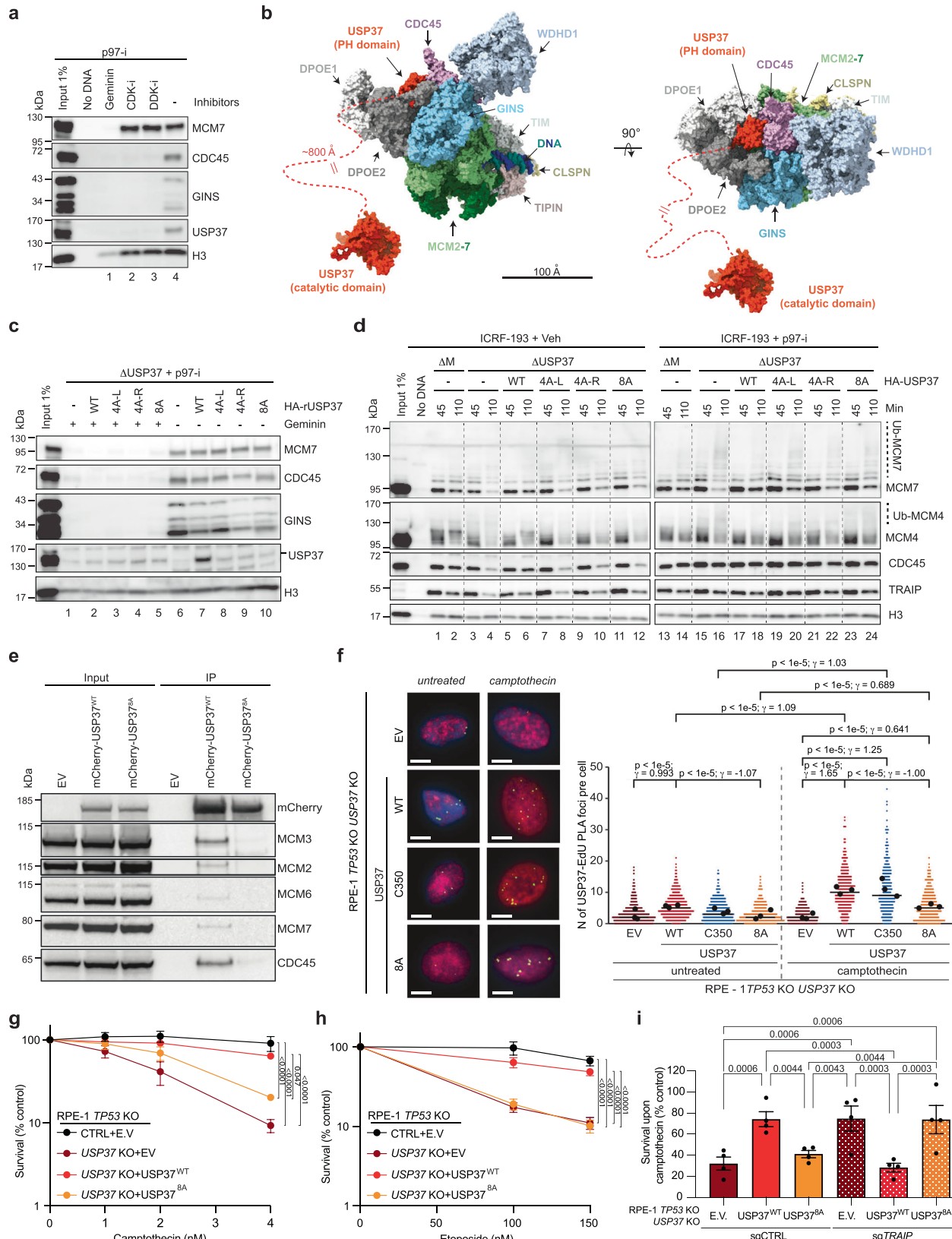

human and mouse cells[49]. Consistent with this interpretation, USP37 did not induce deubiquitylation of CMGs stalled on the outer edges of a 1 kb LacR array, which would probably disrupt 3D chromatin architecture and might also prevent physical coupling of replisomes. Future experiments will be needed to address how TRAIP promotes ubiquitylation of distant replisomes in the absence of USP37.

Unlike CMGs stalled at long range, replicative helicases stalled at ICLs undergo extensive ubiquitylation by TRAIP even in the presence of USP37[11]. Thus, an important question is how TRAIP overcomes the inhibitory activity of USP37 in this setting? We envision two explanations: first, TRAIP activity is dramatically higher at close range, such that it overwhelms USP37 activity at converged CMGs; second, USP37

**Fig. 5 | Predicted CDC45-USP37 interaction surface is important for USP37 function at the replisome. a** *Xenopus* sperm chromatin was incubated in egg extracts with indicated inhibitors for 10 min, recovered and blotted. See also Supplementary Fig. 10b. **b** Right, the predicted structure in Supplementary Fig. 10f aligned to the structure of human CMG (PDB:7pfo). Left, the same model rotated 90°. Orange dashed line, disordered region. **c** *Xenopus* sperm chromatin was incubated in egg extracts supplemented with WGE-expressed recombinant USP37 variants, recovered and blotted. Geminin was added where indicated to block replication. See also Supplementary Fig. 11a, b. **d** Plasmid DNA was incubated in the indicated egg extracts with different WGE-expressed recombinant USP37 variants, ICRF-193 and p97-i as indicated, recovered and blotted. Samples are from the same experiment, blots processed in parallel. See also Supplementary Fig. 11c, d. Abbreviations as in Supplementary Fig. 10k. **e** Extracts of HEK293T cells expressing mCherry (empty vector, EV), mCherry-USP37^WT, or mCherry-USP37^8A were subjected to mCherry immunoprecipitation (IP) followed by immunoblotting. *n* = 3 independent experiments. **f** Representative images of USP37-EdU PLA. Scale bar: 10 μm (left); Dot plot indicating the number of PLA foci between mCherry-tagged

EV, USP37^WT, USP37^C350, or USP37^8A and EdU (replicating DNA) in untreated- or camptothecin-treated cells (right). Untreated: *n* = 291 (EV), 329 (WT), 307 (C350), 287 (8A) cells; camptothecin: *n* = 197 (EV), 320 (WT), 287 (C350), 300 (8A) cells. Bar, median of three independent experiments. Black dots, means in each biological replicates. Statistics reflect two-tailed Wilcoxon rank-sum tests for *p* values without adjustment for multiple hypothesis testing and Cohen's d for effect sizes (γ). **g, h** Clonogenic survival assays of control (CTRL) or *USP37* KO (clone 10) cells expressing mCherry (EV), mCherry-USP37^WT, or mCherry-USP37^8A treated with camptothecin (**g**) or etoposide (**h**). CTRL + EV and USP37 KO+ mCherry-USP37^WT samples are as in Fig. 1h, i. *n* = 3 independent experiments. Bars represent means ± SEM. Statistical analysis for (**g, h**) with two-way ANOVA and Šídák test for multiple comparisons. **i** Clonogenic survival assays of *USP37* KO cells expressing mCherry (EV), mCherry-USP37^WT, or mCherry-USP37^8A transduced with a control or *TRAIP*-targeting sgRNA upon treatment with camptothecin. Bars, means ± SEM. Statistical analysis with two-way ANOVA and Holm-Šídák test for multiple comparisons. Source data provided as a Source Data file.

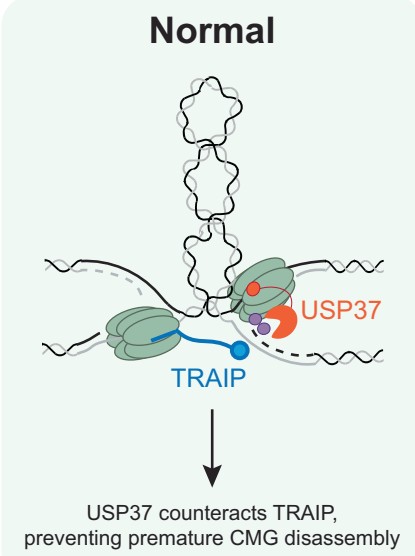 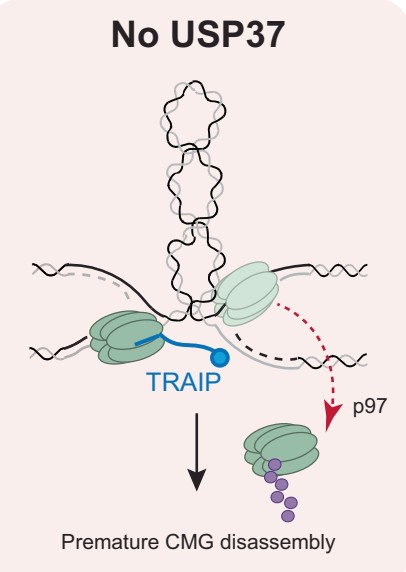 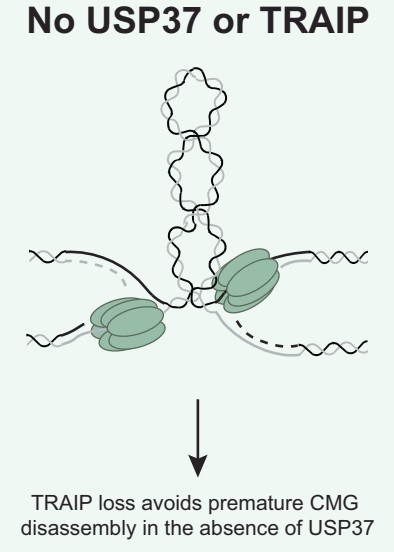

**Normal**

**No USP37**

**No USP37 or TRAIP**

USP37 counteracts TRAIP, preventing premature CMG disassembly

Premature CMG disassembly

TRAIP loss avoids premature CMG disassembly in the absence of USP37

**Fig. 6 | Model of USP37's role in protecting replisomes from premature disassembly.** In wild-type cells (WT), USP37 counteracts *trans* ubiquitylation by TRAIP (Supplementary Fig. 1a) when CMGs are stalled at sites of DPCs/topological stress. Thus, USP37 activity allows replisomes to eventually complete DNA synthesis and be unloaded by a termination-specific CRL2^LRR1-dependent pathway. In the absence of USP37, TRAIP hyper-ubiquitylates CMG at sites of topological stress, causing premature CMG disassembly, DNA damage, and cell death. When both USP37 and TRAIP are absent converging CMGs cannot be prematurely ubiquitylated and, thus, may ultimately terminate normally. For simplicity, the region in between two stalled replisomes is shown as plectonemes.

is negatively regulated when forks meet on either side of an ICL. More work will be required to unravel how the balance of TRAIP and USP37 is regulated under different conditions.

Although aphidicolin induces DNA damage and cell death in *USP37* knockout cells, TRAIP loss did not rescue these phenotypes, suggesting that USP37 counteracts the activity of another E3 ubiquitin ligase in this setting. In line with this observation, we found that TRAIP loss on its own did not hypersensitize cells to aphidicolin, and aphidicolin induced TRAIP dissociation from chromatin in egg extracts (Supplementary Fig. 5e). As such, TRAIP is not available to ubiquitylate proteins at the replication fork during this form of stress. As the only other factor known to ubiquitylate CMG, CRL2^LRR1 is a strong candidate for the ligase counteracted by USP37 under such circumstances[55].

Together with previous studies[7,36,37], our findings strongly suggest that USP37 binds the replisome directly. Consistent with this idea, we found that USP37 binding to chromatin requires DNA licensing and the activities of the DDK and CDK kinases. Moreover, AlphaFold made strong reciprocal predictions that USP37 and CDC45 interact, highlighting CDC45 as a likely tethering point for USP37 at the replisome.

Accordingly, mutations in the USP37 PH domain predicted to disrupt CDC45 binding impaired the association of USP37 with replicating chromatin and its ability to deubiquitylate stalled CMGs in egg extracts. Additionally, this USP37 mutant failed to co-immunoprecipitate with CMG, and it failed to protect cells from TOP1 and TOP2 poisons. We speculate that the presence of a long flexible linker between the PH and catalytic domains of USP37 allows this DUB to deubiquitylate multiple proteins at the fork (Fig. 5b). Consistent with this, we observed enhanced ubiquitylation of multiple CMG subunits upon USP37 loss (MCM7, MCM4, and MCM6). Interestingly, association of USP37^WT but not USP37^8A with replicating DNA increased in cells treated with camptothecin, suggesting USP37's interaction with CDC45 may increase in the presence of stress to preserve stressed replisomes.

Several observations suggest that USP37 might also regulate CMG unloading in the absence of stress. Specifically, in the absence of genotoxic agents, USP37 knockout cells exhibit modestly elevated levels of γH2AX foci and decreased association of CDC45 with nascent DNA, both of which were partially rescued by TRAIP loss (Fig. 2h;

Fig. 3d). One interpretation is that USP37 counters TRAIP-dependent CMG ubiquitylation at long replicons or at endogenous barriers that inhibit termination. Interestingly, we and others also observed that USP37 loss enhances CMG ubiquitylation by CRL2[Lrr1] (Fig. 4c, lanes 1–2; see also refs. 55,56). However, because CRL2[Lrr1] is thought to be recruited to CMGs that have completed DNA replication, this hyper-ubiquitylation is unlikely to be deleterious[8,9]. Therefore, we propose that cell proliferation defects observed in untreated cells lacking USP37 are most likely due to unrestrained TRAIP activity[10,55,56].

In conclusion, our findings identify USP37 as a regulator of genome integrity that prevents premature disassembly of stressed replisomes by TRAIP. Furthermore, since the catalytic activity of USP37 is essential to protect against topoisomerase inhibitors and replication-stalling agents, our work suggests that USP37 inhibitors might be selectively toxic to cancer cells exhibiting elevated replication stress. In addition, our results suggest that inhibiting USP37 might ameliorate phenotypes associated with microcephalic dwarfism that is caused by *TRAIP* mutations[54].

## Methods

### Animal ethics
Egg extracts were prepared using adult female *Xenopus laevis* (Nasco Cat #LM0053MX). All experiments involving animals were approved by the Harvard Medical Area Standing Committee on Animals (HMA IACUC Study ID IS00000051-6, approved 10/23/2020, and IS00000051-9, approved 10/23/2023). The Harvard Medical School has an approved Animal Welfare Assurance (D16-00270) from the NIH Office of Laboratory Animal Welfare.

### Preparation of DNA constructs
DNA replication experiments in Figs. 3, 5d and Supplementary Fig. 5 were performed using the pKV45 plasmid[10]. DNA replication experiments in Fig. 4d, e and Supplementary Fig. 9c, were performed using the pJD156 plasmid[50]. To generate a panel of pmeDPC plasmids, we first generated new variants of the pJLS3 plasmid[48]. To this end, we first removed 48xlacO array by digesting plasmid with BsrGI-HF (NEB) and PstI-HF (NEB), followed by blunting and ligation of the digested plasmid using Quick Blunting and Quick Ligation kits (NEB). For pJLS3-56, the resulting pJLS3-no_lacO was first digested with BamHI-HF (NEB) and ApoI-HF (NEB), blunted and ligated using Quick Blunting and Quick Ligation kits (NEB), respectively. The resulting pJLS3-61 was then mutagenized to substitute two nt.BbvCI nicking sites with nt.BspQI sites and to reduce the distance between M.HpaII sites using OK109/OK110 primers (Supplementary Table 1) and a Q5 Site-Directed Mutagenesis Kit (NEB). To generate pJLS3-305, a gene block containing one copy of the human ubiquitin gene was introduced into a BssHI/ApoI-digested pJLS3-no_lacO using NEBuilder HiFi DNA Assembly Cloning Kit (NEB). The pJLS3-1033 plasmid was generated by insertion of three additional copies of the human ubiquitin gene into pJLS3-305 in a stepwise manner. pJLS3-305 was first PCR-amplified using OK113/OK114 primers (Supplementary Table 1), and the second copy of the human ubiquitin gene with a 3′ 15-nt spacer (complementary to OK115 primer) was then introduced into the resulting PCR fragment using NEBuilder HiFi DNA Assembly Cloning Kit (NEB). The resulted pJLS3-562 was first PCR amplified using OK113/OK115 primers (Supplementary Table 1), the third copy of the human ubiquitin gene with a 3′ 15-nt spacer (complementary to OK117 primer) was then introduced into the resulted PCR fragment using NEBuilder HiFi DNA Assembly Cloning Kit (NEB). To generate pJLS3-1033, the resulted pJLS3-805 was first PCR amplified using OK113/OK117 primers (Supplementary Table 1), and the fourth copy of the human ubiquitin gene was cloned into the resulted backbone using NEBuilder HiFi DNA Assembly Cloning Kit (NEB). To generate the pmeDPC-56 substrate, the plasmid was first nicked with nt.BbvCI (NEB) and ligated with "Dual-Top/Top-nt.BbvCI" oligonucleotide containing fluorinated cytosines (5′-TCAGCATC[C5-fluor dC]

GGTAGCTACTCAATC[C5-fluor dC]GGTACC-3′). The resulted plasmids were then nicked with nt.BspQI (NEB) and ligated with "Dual-Top/Top-nt.BspQI" oligonucleotide containing fluorinated cytosines (5′- CAG-CATC[C5-fluor dC]GGTAGCTACTCAATC[C5-fluor dC]GGCTCTTCA-3′), gel purified, and subsequently crosslinked to methylated M.HpaII-His_6 (kind gift of Justin Sparks) in reaction buffer (50 mM Tris-HCl, pH 7.5, 5 mM 2-mercaptoethanol, 10 mM EDTA) supplemented with 100 μM S-adenosylmethionine (NEB) at 37 °C overnight[57]. All other substrates were generated as described above, but both strands were nicked with nt.BbvCI (NEB) and ligated with "Dual-Top/Top-nt.BbvCI" oligonucleotide containing fluorinated cytosines (5′-TCAGCATC[C5-fluor dC]GGTAGCTACTCAATC[C5-fluor dC]GGTACC-3′).

### DNA replication using egg extracts
*Xenopus* egg extracts and de-membranated sperm chromatin were prepared as described[58]. To carry out DNA replication in egg extracts, licensing was first performed by supplementing a high-speed supernatant (HSS) of egg cytoplasmic fraction with plasmid DNA at a final concentration of 7.5 ng/μL (for nascent strand analysis), 15 ng/μL (plasmid pull-down assay) or sperm chromatin at a final concentration of 10,000 sperm/μL. For replication of LacR-bound plasmids, pJD156 plasmid DNA (300 ng/μL) was mixed with an equal volume of 60 μM LacR[50] and incubated for 60 min at room temperature prior to addition into HSS. Licensing was allowed to proceed for 30 min at room temperature. To inhibit licensing, HSS was preincubated for 10 min at room temperature with Geminin, at a final concentration of 10 μM. To initiate replication, 2 volumes of 50% nucleoplasmic extract (NPE) diluted with 1xELB-sucrose (10 mM Hepes-KOH pH 7.7, 2.5 mM MgCl_2, 50 mM KCl, 250 mM sucrose) was added to 1 volume of licensing reaction. To inhibit CMG assembly, NPE was supplemented with 50 μg/mL recombinant GST-p27[Kip] ("CDK-i") or 50 μM PHA-767491 (Sigma-Aldrich PZ0178, "DDK-i") and preincubated at room temperature for 15 min prior to addition to licensing reaction. To inhibit TOPIIα, replication reactions were supplemented with ICRF-193 (Sigma-Aldrich, I4659-1MG) to a final concentration of 200 μM at 5 min after NPE addition. To prevent nascent strand synthesis in Supplementary Fig. 5e, replication reactions were supplemented with 50 ng/μL of aphidicolin (Sigma-Aldrich) at 5 min after NPE addition. For NMS-873 ("p97-i"; Sigma-Aldrich, SML1128-5MG) treatment, replication reactions were supplemented with NMS-873 to a final concentration of 200 μM at 40 mins after NPE addition or it was added 5 min prior to replication initiation into NPE (Fig. 4c). For MLN-4924 ("Cul-i"; Active Biochem, A-1139) treatment, NPE was supplemented with MLN-4924 to a final concentration of 400 μM in the replication reaction 5 min prior to replication initiation (Fig. 4c). In Fig. 4d, e, replication reactions where indicated were supplemented with Cyclin B-Cdk1 (Sigma-Aldrich, 14–450 M) to a final concentration of 50 ng/μl at 30 min.

For nascent strand analysis in Supplementary Figs. 5b, 9c, replication reactions were supplemented with 0.16 mCi/mL of [a-32P]dATP (Revvity, BLU512H500UC). At the indicated points, samples of the replication reactions were quenched in five volumes of replication stop buffer (80 mM Tris-HCl pH 8.0, 8 mM EDTA, 0.13% phosphoric acid, 10% Ficoll 400, 5% SDS, 0.2% bromophenol blue). Proteins in each sample were digested with 20 mg of Proteinase K (Roche 3115879001) for 1.5 h at 37 °C, and the samples were then resolved on the native 0.8% agarose gel. The dried gels were imaged on the Typhoon FLA 700 PhosphorImager (GE Healthcare).

### Immunodepletions and rescue experiments in egg extracts
For all immunodepletions, 0.5 volumes of the 1 mg/mL affinity-purified antibodies were pre-bound to 1 volume of Dynabeads Protein A (Invitrogen 10002D) by gently rotating at 4 °C overnight. 1.5 volumes of undiluted HSS or 50% NPE diluted with 1xELB were immunodepleted by three rounds of 1-h incubation with 1 volume of antibody-bound Dynabeads at 4 °C. For rescue experiments, one

volume of immunodepleted NPE was supplemented with 0.05 volume of wheat germ extract expressing HA-USP37, TRAIP, or wheat germ extract containing an empty vector.

### SDS-PAGE analysis and western blotting in egg extracts

Protein samples were diluted with SDS Sample Buffer to a final concentration of 50 mM Tris [pH 6.8], 2% SDS, 0.1% Bromophenol blue, 10% glycerol, and 5% β-mercaptoethanol and resolved on Mini-PROTEAN or CRITERION precast gels (Bio-Rad) or home-made 6% polyacrylamide gels (MCM7, CDC45). Gels were then transferred to PVDF membranes (Thermo Scientific, PI88518). Membranes were blocked in 5% nonfat milk in 1x PBST for 1 h at room temperature, then washed three times with 1x PBST, and incubated with primary antibodies diluted to 1:300–1:12,000 in 1x PBST containing 1% BSA overnight at 4 °C. Following washes with 1x PBST, membranes were incubated for 1 h at room temperature with goat anti-rabbit horseradish peroxidase-conjugated antibodies (Jackson ImmunoResearch) at 1:10,000–1:30,000 dilution, light chain-specific mouse anti-rabbit antibodies (Jackson ImmunoResearch) at 1:10,000 dilution, or rabbit anti-mouse horseradish peroxidase-conjugated antibodies (Jackson ImmunoResearch) at 1:2000 dilution, diluted in 5% nonfat milk in 1x PBST. Membranes were then washed three times with 1x PBST, developed with SuperSignal West Dura substrate (Thermo Fisher), and imaged using an Amersham ImageQuant 800 (Cytiva). Western blots in Supplementary Fig. 10k, l were quantified using a band-it tool (https://thecodingbiologist.com/posts/band-it-the-gel-band-quantifier).

### Antibodies used for western blotting of *Xenopus* proteins

The following rabbit polyclonal antibodies were used for Western blotting: MCM6 (1:5000; ref. 7); MCM7 (1:12,000) and RPA (1:7000; ref. 59; MCM4 (1:2000; Bethyl, A300-193A); CDC45 (1:20,000; ref. 60); H3 (1:500; Cell Signaling, 9715S); TRAIP (1:10,000) and GINS (1:5000) (ref. 11); SMC3 (1:5000; ref. 61); SMC2 (1:5000; ref. 21); USP37.L (1:5000; this study); TOP2α (1:5000; ref. 50); RTEL1-N (1:2500; ref. 48); P-H3 (S10) (1:1000; Cell Signaling, 9701S); UBXN7 (1:5000; ref. 62). Mouse monoclonal antibodies against the following proteins were used for western blotting: ubiquitin (1:300; Santa Cruz Biotechnology, sc-8017 P4D1). Rabbit polyclonal antibodies raised against the C terminus of *Xenopus laevis* Usp37.L (Ac-CSQPVSTELNWPTRPPL-OH) were prepared by BioSynth.

### Plasmid pulldown from egg extracts

Plasmid pull-down was performed as described previously[51]. Briefly, 4 µl of streptavidin-coated magnetic beads (Dynabeads M-280, Invitrogen) were incubated with 8 pmol of biotinylated LacR in 6 volumes of LacR-binding buffer (50 mM Tris-HCl [pH 7.5], 150 mM NaCl, 1 mM EDTA [pH 8.0], 0.02% Tween-20) for 40 min at room temperature. The beads were then washed three times with 5 volumes of 20 mM HEPES-KOH [pH 7.7], 100 mM KCl, 5 mM MgCl$_2$, 250 mM sucrose, 0.25 mg/ml BSA, and 0.02% Tween-20, resuspended in 40 µl of the same buffer, and kept on ice. At indicated time points, 6 µl of replication reactions were mixed with the prepared beads and rotated for 30 min at 4 °C. The beads were then washed three times with 200 µl of 20 mM HEPES-KOH [pH 7.7], 100 mM KCl, 5 mM MgCl$_2$, 0.25 mg/ml BSA, and 0.03% Tween-20. Following complete removal of the washing buffer, beads were mixed with 20 µl of 1xSDS Sample buffer and boiled at 95 °C for 2 min.

### Sperm chromatin spin-down from egg extracts

Sperm chromatin spin-down was performed as previously described[63]. Briefly, 10 min after replication initiation, 12 µl of replication reaction was mixed with 60 µl of ice-cold 1xELB + 0.2% Triton X-100, and chromatin and associated proteins were isolated by centrifugation through a 180 µl sucrose cushion. 1xELB + 0.5 M Sucrose). Pelleted chromatin was washed twice with ice-cold 200 µl of 1xELB, resuspended in 2x SDS sample buffer (100 mM Tris pH 6.8, 4% SDS, 0.2%

bromophenol blue, 20% glycerol, 10% β-mercaptoethanol). Chromatin and associated proteins were then solubilized by two rounds of sequential boiling at 95 °C for 7 min and vortexing for 10 s prior to western blotting. For Supplementary Fig. 6i, k, replication reactions were supplemented with 200 µM ICRF-193 at 5 min after replication initiation.

### Expression of proteins in wheat germ protein expression system

The wild-type *Xenopus* HA-USP37 open reading frame (ORF) was cloned by introducing the respective gBlocks (IDT) using NEBuilder HiFi DNA Assembly Cloning Kit (NEB) into the AsiSI (NEB) and EcoRI-HF (NEB) double-cut pF3A WG (BYDV) Flexi vector (Promega). The *Xenopus* TRAIP ORF was cloned as above, but the pF3A WG (BYDV) Flexi vector (Promega) was cut with PvuI-HF (NEB) and EcoRI-HF (NEB). The C347S mutation was introduced into the pF3A-HA-USP37 vector using "Round-the-Horn" and primers OK74/OK75 (Supplementary Table 1). To introduce R11A, N13A, K21A, W22A, K63A, R71A, D85A, K86A (8 A) mutations into the USP37 gene, pF3A-HA-USP37 vector was first amplified using OK99/OK100 primers (Supplementary Table 1), and the PH-8A gBlock (IDT) (Supplementary Table 1) corresponding to 10–285 nt of the USP37 gene (IDT) and containing the indicated mutations was introduced using NEBuilder HiFi DNA Assembly Cloning Kit (NEB). For protein expression, 3 volumes of TnT® SP6 High-Yield Wheat Germ Protein Expression System (Promega) was incubated with 2 volumes of 100 ng/µL purified plasmid at 25 °C for 2 h and used immediately.

### In vitro deubiquitylation assay

About 10 ul of wheat germ reactions expressing HA-USP37 variants were incubated with 25 µl of Pierce™ Anti-HA Magnetic Beads (Thermo Fisher) equilibrated in 2xELB-sucrose-BSA-NP-40 buffer (20 mM HEPES-KOH [pH 7.7], 100 mM KCl, 5 mM MgCl$_2$, 250 mM sucrose) for 1 h at 4 °C, washed five times with 2xELB-sucrose-BSA-NP-40 + 5 mM DTT. The beads were then resuspended with 30 µl of 1 uM K48-linked tetraubiquitin (LifeSensors) diluted with 2xELB-sucrose-BSA-NP-40 + 5 mM DTT and rotated on the wheel at room temperature. At indicated time points, 5 µl of the reactions was mixed with 2x SDS sample buffer (100 mM Tris pH 6.8, 4% SDS, 0.2% bromophenol blue, 20% glycerol, 10% β-mercaptoethanol), and boiled at 95 °C for 2 min prior to western blotting.

### AF-M prediction generation

Unless stated otherwise all structure predictions were generated using AlphaFold-multimer v2.3 (AF-M)[64] via ColabFold 1.52 (ref. 65). AF-M was run with v2.3 weights1 ensemble, 3 recycles, templates enabled, dropout disabled, and maximum Multiple Sequence Alignments (MSA) depth settings (max_seq = 508, max_extra_seq = 2048). MSAs (paired and unpaired) were fetched from a remote server via the MMSeq2 API[66] that is integrated into the ColabFold pipeline. To reduce computational burden during the interaction screens, we ran three out of the five AF-M v2.3 models (1,2, and 4) and did not run any pairs where the total amino acid length exceeded 3,600 residues (GPU memory limit).

For Supplementary Fig. 10h, i, five predictions were generated by folding *Human* or *Xenopus* CDC45-USP37 pairs, respectively, using AF-M v2.3 as described above, but with 12 recycles. The top-ranked models were then relaxed using AMBER (https://ambermd.org/index.php), and hydrogen atoms were removed with ChimeraX1.8. H bonds and salt bridges were visualized with UCSF ChimeraX1.8 using 3 Å distance tolerance and 20° angle tolerance criteria. For simplicity, only key residues predicted to interact are shown and/or labeled.

### AF-M prediction analysis

Python scripts were utilized to analyze structural predictions generated by AF-M as previously described[53,67]. Briefly, confident

interchain residue contacts were extracted from AF-M structures by identifying proximal residues (heavy atom distance <5 Å) where both residues have pLDDT values >50 and PAE score <15 Å. All downstream analysis of interface statistics (average pLDDT, average PAE) were calculated on data from these selected inter-residue pairs (contacts). Average interface pLDDT values above 70 are generally considered confident[68]. The average models score was calculated by averaging the number of independent AF-M models that predicted a specific inter-residue contact across all unique pairs in all models. This number was additionally normalized by dividing by the number of models run to produce a final average model score that ranges from 0 (worst) to 1 (best). An average model value above 0.5 is generally considered confident. pDockQ estimates of interface accuracy scores were calculated independently of the contact analysis described above using code that was adapted from the original implementation as described in ref. 69 pDockQ values above 0.23 are considered confident.

## SPOC interaction analysis

The random forest classifier algorithm (structure prediction and omics classifier) SPOC was used to score the binary interaction predicted by AF-M as described in ref. 53. Briefly, this classifier was trained to distinguish biologically relevant interacting protein pairs from non-relevant interaction pairs in AF-M interaction screens. SPOC assigns each interaction a score that ranges from 0 (worst) to 1 (best). Higher scores indicate that AF-M interface metrics and several types of externally omics datasets are consistent with the existence of the binary interaction produced by AF-M. The highest confident interactions are generally associated with SPOC scores above 0.5.

## Cell culture and cell line generation

RPE-1 and HEK293T cells were originally obtained from Prof. Jonathon Pines. RPE-1 *TP53*[KO] (ref. 70) were cultured in Dulbecco's Modified Eagle Medium: Nutrient Mixture Ham's F-12 (DMEM/F-12, Sigma-Aldrich). U2OS Cas9 (ref. 71), HEK293T, and HEK293T Lenti-X (Takara Bio) cells were cultured in DMEM (PAN-biotech or Sigma-Aldrich). All cell lines were cultured at 37 °C and 5% $CO_2$. All media were supplemented with 10% (v/v) fetal bovine serum (FBS, BioSera), 100 U/mL penicillin, 100 µg/mL streptomycin (Sigma-Aldrich), 2 mM L-glutamine. 10 µg/mL blasticidin (Sigma-Aldrich) was used to select for Cas9-expressing cells. RPE-1 cells stably expressing the mCherry-USP37 vectors were selected with 1 µg/mL G418. USP37 knock-out cells were generated by transient transfection of the sgRNA: AATGTGGTGCTTCGACCCAG targeting exon 5. Three or four days after, single cells were plated, and colonies were picked when visible. Editing was analysed by TIDE (Netherlands Cancer Institute, http://shinyapps.datacurators.nl/tide/) on DNA amplified using the following primers: TGGTCTGTAGTCTAGTCATAGCCT, CCCTTGGTGCAAGATCTCTGT. TRAIP knock-out cells were generated by transduction of the LentiGuide-NLS-EGFP-Puro (kind gift from Durocher laboratory) containing the GACGTGGCCGCCATCCACTG gRNA cloned using the following primers: CACCGACGTGGCCGCCATCCACTG, AAACCAGTGGATGGCGGCCACGTC. Editing was analysed by TIDE on DNA amplified using the following primers: TTGCCCAGGC-TAACGGTTTT, AGGCGAAGTATTCACGCTCC.

## CRISPR/Cas9 screens

For CRISPR/Cas9 screens, U2OS Cas9 wild-type and *USP37* knockout cells were transduced at an MOI of 0.3 and 500-fold coverage of the Brunello library[38]. Afterward, transductants were selected with 1.5 µg/mL puromycin for 12 days. Library preparation and next-generation sequencing of the samples was performed as described previously[72]. Guide-enrichment analysis was performed using DrugZ [73].

## Clonogenic survival assays

For clonogenic assays, 500 RPE-1 or U2OS cells were plated in six- well plates, treated with drugs 24 h later, and stained and counted 7–10 days later, when visible colonies were formed. Cells were treated with etoposide (VWR International Ltd., Cat# CAYM12092-500), camptothecin (C9911, Sigma-Aldrich), talazoparib (Stratech Scientific, S7048-SEL), ICRF-193 (GR332, BIOMOL International), aphidicolin (Merck A0781), or hydroxyurea (H8627, Sigma-Aldrich).

## Competitive cell-growth assay

Control or *USP37* knockout human U2OS cells were transfected with sgRNA targeting TRAIP (GACGTGGCCGCCATCCACTG). Samples were collected on the indicated days and subcultured (to prevent confluency) into fresh medium. To calculate the percentage of indel formation, DNA was extracted and regions near the cut-site were PCR-amplified with the indicated primers (Fw: TTGCCCAGGC-TAACGGTTTT, Rev: AGGCGAAGTATTCACGCTCC), sequenced, and analyzed by TIDE (Netherlands Cancer Institute, http://shinyapps.datacurators.nl/tide/).

## Plasmid, siRNA and viral transduction

The plasmids were obtained from VectorBuilder (mCherry, VB230126-1146bke; USP37-WT, VB230119-1435kfc; USP37-C350, VB230126-1139fpu; USP37-8A, VB230119-1445ecf). To generate stable cells, the virus was first produced in Lenti-X 293T cells by co-transfecting the packaging constructs psPAX2 (Addgene #12260) and pMD2.G (Addgene #12259) with the plasmid of interest using TransIT-LT1 (Mirus Bio) according to the manufacturer's protocol. Viral supernatant was then incubated with cells in the presence of Polybrene (10 µg/mL) (Merck), followed by positive selection with Geneticin (Gibco). Transient transfection of Hek293T was obtained using was carried using TransIT-LT1 (Mirus Bio) according to the manufacturer's protocol. siRNA transfection was performed using Lipofectamine RNAiMAX (Thermo Fisher Scientific) according to the manufacturer's protocol. Cells were seeded for the experiment the following day and were treated and fixed the day after plating. Total RNA was extracted using the Maxwell® RSC simplyRNA Tissue Kit (Promega) and reverse-transcribed into cDNA using the SuperScript™ VILO™ Reverse Transcriptase (Thermo Fisher Scientific), following the manufacturer's instructions. qPCR was performed using Fast SYBR™ Green Master Mix (Thermo Fisher Scientific).

## Immunoblotting

Cells were lysed in Laemmli buffer (2% sodium dodecyl sulfate (SDS), 10% glycerol, 60 mM Tris-HCl pH 6.8), incubated for 5 min at 95 °C and resolved by SDS-PAGE. After transferring onto a nitrocellulose membrane, membranes were blocked with 5% milk or BSA in TBS-T buffer (Tween-20, 0.1%) and incubated with primary antibodies diluted in 5% milk or BSA in TBS-T. Membranes were washed in TBS-T and probed with secondary antibodies. After secondary antibody incubation, membranes were washed in TBS-T, incubated with enhanced chemiluminescence (ECL) mixture for 5 min in the dark, and developed using films or a Chemidoc imaging system (Bio-Rad).

## Antibodies used for western blotting of *Human* proteins

The following antibodies were used for western blotting of proteins in human cell lysates: USP37 (1:1,000; Abcam, ab72199), mCherry (1:1,000; Abcam, ab167453), MCM2 (1:1,000; Abcam, ab4461), MCM6 (1:1,000; Bethyl, A300-127A), MCM7 (1:1,000; Santa Cruz, sc-9966), CDC45 (1:1,000; Santa Cruz, sc-20685), GAPDH (1:1,000; Millipore, MAB374), TUBULIN (1:1,000; Sigma-Aldrich, T9026).

## Immunoprecipitation

For immunoprecipitation, around 2 million Hek293T cells were plated in 10 cm dishes. The cells were transfected with the indicated plasmids

using MirusLT1 according to the manufacturer's procedure. One day later, cells were washed twice with cold PBS and lysed in 1.5 mL of IP buffer (20 mM Tris-HCL pH 7.5, 150 mM NaCl, 2 mM MgCl$_2$, 10% glycerol, 0.5% NP40, and EDTA-free protease and phosphatase inhibitors) and 10 μL benzonase (Millipore) for 45 min. Lysates were centrifuged at 21,100x$g$ for 10 min and supernatants were incubated with 20 μL RFP trap magnetic beads (Cromoteck) for 2 h at 4 °C. Samples were washed 4x with IP buffer and finally eluted in 40 μL LDS buffer 2x + 1 mM DTT.

## Immunofluorescence

Cells were plated in 24-well imaging plates and treated with the indicated drugs 2 days after in the presence of 10 μM EdU (A10044, Thermo Fisher Scientific). After treatments, cells were pre-extracted in 0.2% Triton X in PBS on ice for 5 min and fixed in 4% paraformaldehyde for 10 min. Cells were then blocked in PBS−5% bovine serum albumin (BSA) for at least 30 min. Click reaction was performed to detect S-phase cells with 10 μM AF647-Azide (Jena Bioscience), 4 mM CuSO$_4$ (Sigma-Aldrich), 10 mM (+)-sodium l-ascorbate (Sigma-Aldrich) in a 50 mM Tris buffer. Cells were then incubated with primary antibodies overnight (RPA, 1:1,000; NA18 Calbiochem, and γH2AX, 1:1,000; 2577 Cell Signaling Technologies), washed in PBS−0.2% Tween three times and incubated with secondary antibodies for 45 min. After three more washes in PBS−0.2% Tween, nuclei were stained with 4′,6-diamidino-2-phenylindole for 10 min. Images were acquired and analyzed using the Opera Phoenix microscope.

## Proximity ligation assay

The PLA assay was performed as previously described[74]. Briefly, cells were labeled with 10 μM EdU for 10 min, permeabilized 5 min in nuclear extraction buffer on ice (10 mM PIPES, 300 mM sucrose, 20 mM NaCl, 3 mM MgCl$_2$, 0.5% Triton X-100), and fixed in 4% PFA for 10 min. When indicated, cells were treated with 1 μM p97-i for 1 h and/or with 100 nM CPT 20 min prior to EdU pulse. The click reaction was performed with the Click-iT Kit (Invitrogen C10337, C10339), replacing the supplied Alexa Fluor Azide with a mixture of 20 μM Biotin-Picolyl-Azide (Sigma-Aldrich 900912) and 1 μM Alexa Fluor Azide and proceeded according to the manufacturer's instructions. Cells were blocked in 3% BSA/PBS for 2 h at room temperature and incubated with primary antibodies overnight at 4 °C. For EdU-Flag PLA, mouse monoclonal anti-biotin (Jackson Immunoresearch, 200-002-211, 1:2000) and rabbit anti-Flag (Sigma-Aldrich F7425, 1:3000) and for EdU-CDC45 PLA mouse monoclonal anti-biotin (Jackson Immunoresearch, 200-002-211, 1:500) and rabbit anti-CDC45 (Cell Signaling Tech #11881, 1:500) were used. After incubation with anti-Mouse MINUS and anti-Rabbit PLUS PLA Probes (Sigma-Aldrich DUO82004, DUO82002), cells were subjected to proximity ligation and detection with DUOLINK detection Kit following the manufacturer's instructions (Sigma-Aldrich DUO92008, DUO92014). Coverslips were mounted with Prolong antifade with Dapi (Invitrogen P36935) and cells were imaged on a Nikon Eclipse Ni microscope in conjunction with Elements v4.5 software (Nikon). PLA foci in EdU-positive cells were quantified for each condition and data were analyzed with Prism software.

## DNA fiber analysis

Cells were pulse-labeled with 25 μM CldU for 20 min, washed twice with 250 μM IdU with or without 100 nM camptothecin, cells where then pulsed with 250 μM IdU with or without 100 nM camptothecin 20 min before harvesting. For replication fork restart assays 100 nM camptothecin was added to the initial 25 μM CldU pulse and washed off with two washes of 250 μM IdU containing media before being pulsed with 250 μM IdU and then harvesting. DNA fiber analysis was carried out as previously described[75]. Images were captured on a Nikon Eclipse Ni microscope and analysed using ImageJ v1.48.

## Figures and schematics

Figures were made with Adobe Illustrator 2021. All figures with AF-M predictions were generated in UCSF ChimeraX1.8.

## Statistics and reproducibility

Graphs and statistical tests were made using Prism v9 or Prism v10.4.1 (GraphPad Software). For immunofluorescence data in Figs. 1g, h, 2g, h, 3d, 5f, graphs and statistical tests were made using R version 4.2.3. Statistics indicate two-tailed Wilcoxon rank-sum tests for $p$ values without adjustment for multiple hypothesis testing. Observations were stacked across replicates. Effect size (γ) was calculated using Cohen's d. The mean increase is generally classified as small, medium, and large when γ = 0.2, γ = 0.5, and γ ≥ 0.8, respectively. The mean decrease was classified as small when γ = −0.2, γ = −0.5, and γ ≤ −0.8, respectively. Black dots indicate means in each independent experiment; horizontal bars indicate medians. Outliers in Fig. 3d were removed with the ROUT method using Prism v9. For clonogenic survival assays in Figs. 1b−e, i−k, 2c−e, 5g, h statistics was calculated using two-way-ANOVA. Statistics in Supplementary Figs. 4e, 7a, 12a, e, f was calculated using the unpaired nonparametric Mann−Whitney test. Statistics in Supplementary Fig. 7b was calculated using an unpaired $t$-test. For Supplementary Fig. 10l, statistics was calculated using two-way ANOVA (followed by Dunnett post hoc test). $P$ values were considered statistically significant at <0.05. All experiments in *Xenopus* egg extracts were performed independently three times, with the exception of experiments shown in Supplementary Figs. 5b, 9c, which were performed twice as they are a confirmation of previously published results[6,12,58].

## Reporting summary

Further information on research design is available in the Nature Portfolio Reporting Summary linked to this article.

## Data availability

Standardized datasets for Figs. 1a, 2a have been deposited on Dryad (https://datadryad.org/stash/share/uIxGpIs0P4mzIA8Cn9PEecC22 u8A0qLK3DawMu2bgns). Source data are provided with this paper.

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

## Acknowledgements

We thank James Dewar, Karim Labib, and members of the Walter and Jackson laboratories for constructive advice on the manuscript. We also acknowledge all members of the Cancer Research UK Cambridge Institute facilities for their assistance. J.C.W. is supported by NIH grant HL098316. J.C.W. is a Howard Hughes Medical Institute Investigator and an American Cancer Society Research Professor. Research in the S.P.J laboratory is supported by Cancer Research UK (CRUK) Discovery Award DRCPGM\100005, CRUK core grant A:29580, and ERC Synergy Award 855741 (DDREAMM). S.P.J. is an employee of the University of Cambridge. G.D'A and A.V. are funded by an ERC Synergy Award 855741. This project has received funding from AIRC and the European Union's Horizon 2020 research and innovation program under Marie Sklodowska-Curie grant agreement no. 800924 to G.D.A. Research in the G.S.S. laboratory is supported by a CRUK program grant (C17183/A23303).

## Author contributions

O.V.K. performed all experiments in *Xenopus* egg extracts and also generated Fig. 5b and Supplementary Fig. 10e–i. G.D'A. performed experiments in Figs. 1, 2, 5e, g, h and Supplementary Figs. 2, 3, 4a–d, 12b, c, with the help of A.V., D.P., and S.L.R. A.V. and S.L.R. performed experiment in Fig. 5i and 12d and G.D'A. analyzed them. C.J.C. contributed to experimental work on USP37 mode of action, reviewing and editing of the manuscript. N.G. assisted with human cell line generation and characterization. V.G. and S.L. contributed to the design and analysis of datasets in Figs. 1a, 2a. S.L. plotted the data in Figs. 1h, g, 2h, g, 3d, 5f. AF-M predictions were performed by E.S. and O.V.K. Large-scale prediction screening in Supplementary Data 1 and analysis were performed by E.S. Figures 3d, 4e, 5f and Supplementary Figs. 7, 12a, e, f were generated by M.R.G. and S.S.J. under the supervision of G.S.S. R.A.W. generated pAW18 vector and validated recombinant TRAIP activity in egg extracts. O.V.K., G.D'A., J.C.W., and S.P.J. wrote the manuscript. D.P. and E.S. contributed equally to the work.

## Competing interests

J.C.W. is a co-founder of MOMA Therapeutics, in which he has a financial interest. S.P.J. is Chief Research Officer (part-time) at Insmed Innovation UK, Ltd. and founding partner of Ahren Innovation Capital LLP. He is a board member of Mission Therapeutics Ltd. and is a consultant and shareholder of Inflex Ltd. The remaining authors declare no competing interests.
