## [Transparent Peer Review file · Nature Communications]

USP37 prevents premature disassembly of stressed replisomes by TRAIIP

Corresponding Author: Dr Johannes Walter

Parts of this Peer Review File have been redacted as indicated to remove third-party material, and references to personal communication.

Version 0:

Reviewer comments:

Reviewer #1

(Remarks to the Author)

Kochenova et al. found through in vivo and in vitro studies that the deubiquitinase USP37 can protect the CMG complex, preventing its premature dissociation from replication forks under replication stress, thereby maintaining genome stability. The authors performed a CRISPR screen to identify which protein deficiencies cause cellular sensitivity to CPT and identified the deubiquitinase USP37. This result was validated using USP37 knockout monoclonal cells. To further understand why the loss of USP37 sensitizes cells to CPT, the authors conducted another CRISPR screen in USP37 knockout cells and discovered that the loss of the TRAIIP E3 ligase rescues the sensitivity of USP37-deficient cells to CPT, indicating a functional relationship between USP37 and TRAIIP. Previous research in the Walter lab showed that, on interstrand crosslinks, after CMG convergence, TRAIIP promotes CMG dissociation by ubiquitinating CMG. Building on these findings, the authors investigated whether USP37 stabilizes the CMG under CPT-induced replication stress by counteracting TRAIIP and deubiquitinating MCM. They tested this hypothesis using the *Xenopus* egg extract system. The authors further explored the conditions under which TRAIIP ubiquitinates CMG during replication stress and found that approximately 1 kb of distance between two CMGs could trigger CMG ubiquitination, and this triggering mechanism may involve not only the distance but also the chromatin structure between the two CMGs. Finally, the authors used AlphaFold to predict the binding mode and binding sites of USP37 to CMG and performed mutational analysis on these sites. They found that the PH domain of USP37 directly interacts with CDC45. Furthermore, they validated the interaction and function using the *Xenopus* egg extract system. Overall, this study is highly innovative, logically rigorous, well-supported by evidence, and meticulously written.

Major concerns:

1. The authors emphasize that the loss of USP37 causes sensitivity to CPT due to premature CMG dissociation. Please use DNA fiber single-molecule experiments to observe whether replication forks can restart after the removal of CPT and cisplatin treatment.
2. In Figure 3d, the PLA is for CDC45 and EdU. However, since USP37 knockout cells have replication issues, the EdU incorporation might change. The PLA signal should be normalized to the EdU signal. The same applies to Figure 5f.

Minor points:

1. The illustrations in Figures 2D and 2E contain errors.
2. The reference in line 245 needs correction.

Reviewer #2

(Remarks to the Author)

Reviewer #3

(Remarks to the Author)

The scientific work presented here describes the role of USP37 in promoting the deubiquitylation of CMG complexes stalled at replication blocks. This mechanism is crucial to allow the correct execution of DNA replication and prevents genome instability. The manuscript is excellent and uses elegant genetic experiments together with complex in vitro reconstitution assays to derive a clear picture on the role of USP37 and TRAIIP in controlling CMG unloading. The experiments are carefully executed and well-controlled, the literature is discussed in detail and the final model is presented in a crystal-clear manner.

I have minor comments which might help the flow of the manuscript for the non-expert readers.

Line 36 unloading by CRL2Lrr1 (Ref.10). Line 152 Data Fig. 5a; Ref.6).

Authors often use Ref. to indicate References. Is there a particular reason to use this type of format in the above indicated lines?

USP37 is a common essential gene according to DEPMap, although this doesn't subtract from the validity of the reported observations, I believe it is important to clarify whether these cells have completely lost USP37 WT. Alternatively, are experiments conducted in cells shortly after sgRNA mediated gene depletion? Can the authors clarify?

Phenotypes are validated using rescue/put back of USP37 and a catalytically inactive mutant, thus, there is no question that they depend on USP37 but it can be useful to clarify the genetic identity of the clones.

In figure 2E the authors show that Aphidicolin sensitivity is not rescued by double depletion of TRAIIP and USP37. The authors appropriately comment on this in the discussion postulating the presence of a second ligase. Now, in the current position this observation seems to distract rather than enrich the flow. Maybe a rationale for using Aphidicolin in these settings could be briefly mentioned in lines 130-135 (and discussed later).

The results of the two genetic screens (Fig.1A and Fig.2A) conducted could be presented in excel tables to help the reader follow up on parallel observations and increase the impact of the work.

The model presented in Figure 6 is crystal clear but doesn't encapsulate the important message from the manuscript about the ubiquitylation of CMG by TRAIIP in Cis vs Trans.

If the authors refer back to the model in Extended Figure 1A in the Figure legend of Figure 6, can this facilitate the presentation of this important finding ?

Can An explanation of the differential ubiquitination be USP37 unloading from CDC45 based on DNA conformation or fork convergence? This could be one of the reasons why USP37 is not often detected in iPOND and similar techniques?

Reviewer #4

(Remarks to the Author)

CMG (Cdc45-Mcm2-7-GINS) helicase disassembly is a crucial event in DNA replication, playing a key role in maintaining genome integrity by preventing re-replication. A key process regulating CMG disassembly is ubiquitination, largely occurring on the MCM7 subunit of the MCM complex and leading to unfolding by the p97/VCP ATPase. There are two major E3 ubiquitin ligases regulating CMG-MCM7 ubiquitination and disassembly, CRL2LRR1 and TRAIIP. During replication termination, CMGs unloading from chromatin is promoted by CRL2LRR1, which also removes CMGs at specific types of damage. On the other hand, TRAIIP associates with the replisomes and ubiquitinates CMGs at inter-strand cross-links (ICLs), promoting unloading of CMGs by p97. During S phase, TRAIIP apparently ubiquitinates CMGs, and also DNA protein cross-links (DPCs) in trans, and is unable to ubiquitinate the replisome in cis, which is crucial to prevent premature unloading of CMGs. In mitosis, TRAIIP promotes in cis ubiquitination of CMGs and disassembly of the replisome to allow chromosome segregation. Keeping the activity of TRAIIP (and CRL2LRR1) under tight control is crucial to avoid deregulated ubiquitination and disassembly of the CMG complex in interphase.

The present manuscript aims to identify new regulators of DNA topological stress, by a genome wide CRISPR screen in U2OS upon TOP1 inhibitor (CPT). Among the factors whose inactivation causes CPT hypersensitivity, the authors focused their studies on the DUB USP37 by combining cell sensitivity assays in human cells and mechanistic studies in Xenopus egg extracts. The study is interesting and timely, addresses important mechanisms at the basis of the regulation of the CMG complex disassembly and identified a new factor regulating this process.

Overall, the work is well organized, and the results are important. However, in some cases the data obtained were not strong enough to support the conclusions drawn, and there are few important points that need to be addressed to make it competitive for publication in Nature Communications. These key points are discussed below.

Major comments

General

In the manuscript, Xenopus egg extracts have been widely used to measure CMG unloading (Fig. 3-5), and for non-experts of this system it can be difficult to appreciate how it works. It would be helpful to provide more explanation in the main text,

supported perhaps by a schematic drawing, to facilitate understanding of the results.

Furthermore, as several cell lines are used in the study, it would be beneficial to the reader if the cell line used was indicated above a figure.

Another important point, for most of the survival assays, the protein levels of USP37 variants (WT and mutants) are not shown, making it difficult to draw conclusions on the effect of complementation and the relevance of each domain (i.e., RING finger and PH) in USP37 activity.

Also, it is known that USP37 contains 3 ubiquitin interacting motifs (3XUIMs) that mediate the binding to ubiquitin and also regulate its catalytic activity (Manczyk et al, Sci Rep 2019). What is the role of these 3XUIMs in the association with ubiquitinated components of the replisome is missing. For the sake of clarity, the possible contribution of the UIM domains needs to be clarified and tested.

Specific points

Fig. 1h-j: The authors measured the survival of USP37 KO cells and the effect of complementation with USP37 WT and catalytic mutant C350, showing that the C350 mutant does not restore viability of cells upon different types of stress. This experiment is problematic, because the expression of the C350A is very poor compared to WT (as shown in Ext. Fig. 2i) and therefore it is not possible to draw any conclusion from this experiment. To unequivocally prove the need of USP37 enzymatic activity in the survival of cells exposed to genotoxic stress, the experiment should be repeated using cells expressing equal (or at least similar) levels of USP37 WT and C350.

Lines 107-108: 'These data indicated that the catalytic activity of USP37 protects cells from the deleterious effects of replication fork stalling'. Based on the levels of C350 mutant complementation, this cannot be stated.

Lines 99-100: 'these data support a role for USP37 in promoting genome integrity upon treatment with CPT and aphidicolin'. These data do not indicate a role of USP37 in promoting genome integrity but rather in cell survival.

Fig. 3a, Lines 162-163: 'While USP37 depletion did not influence CMG unloading during unperturbed replication....'. Difficult to claim this, since the drop in CMG unloading was very drastic (from 10' to 45'). It would be of help to test this using intermediate timepoints. Also, the difference between lane 9 and 12 is not so convincing (especially for MCM6).

There is a difference between MCM6 and MCM7 ubiquitination after 110 minutes (lane 12). How would the authors explain this?

Fig. 3b: The effect of p97i on the unloading of the fork seems quite small, especially since the blot without the p97i is on another gel, making it difficult to take strong conclusions.

Fig. 5g,h: The authors show that USP37 binds CDC45 and that TRAIIP loss is beneficial to USP37 knockout cells. As the authors state that their data suggest that USP37's interaction with CDC45 counteracts TRAIIP mediated CMG ubiquitylation, it would be nice if the authors include sgTRAIIP conditions.

Minor comments

For the results in Fig.1f,g, Fig.2f,g, Fig. 3d, it would help to see the variability in the assay by showing the median of each replicate in the figure.

Do the USP37-KO cells have a normal cell cycle distribution? Expect lower EdU incorporation for USP37-deficient cells, do the authors also see this in the data from Fig. 1f?

Fig. 1b-d: Why do the RPE1 clones (KO10 and KO19) behave so differently, since the KO looks efficient in both? Does this indicate a clonal variability? Along the same line, why do the U2OS clones (KO6 and KO17; Fig. 1f,g) behave differently in untreated conditions?

Fig. 2f,g: Surprisingly the siTRAIIP treated control cells show less γ H2AX foci and RPA foci upon CPT treatment, however loss of TRAIIP is toxic upon increasing doses of CPT in Fig. 2c. How do the authors explain this?

Fig 3b, line 174: 'Furthermore, this effect was reversed by USP37 WT but not USP37-C347S (Fig. 3b, lanes 14 and 16)'. The effect of the WT complementation is not very strong, the suggestion is to tone it down.

Fig 4d, e: How are the meDPCs induced here?

How can TRAIIP promote ubiquitination at distant CMGs? Not sufficient data in support of this point.

Page 10 lines 263-264: How do you know that in your experiments, MCM7 ubiquitination in mitosis is completely dependent on TRAIIP? It would be advisable to test depletion of TRAIIP in experimental setting as in Fig 4d-e and prove this point. USP37-8A mutant has reduced enzymatic activity compared to WT (Ext fig 7j), which should be mentioned in the text.

Fig. 5a: The authors need to show USP37 levels in the whole cell lysate upon CDKi, as this could affect the expression of USP37 and could therefore also explain the lack of binding to the DNA.

Fig. 5c: Introducing 8 mutations might have quite a dramatic impact on the structure of the PH-domain, with unpredictable effect on the folding of the whole protein. It would be preferable, if possible, to generate a deficient mutant by introducing fewer amino acid alterations.

Fig 5c, lane 8: Again, the levels of expression on WT and 8A are not shown. In the input samples, only 1 lane is shown (and not indicated whether it corresponds to the WT or the 8A sample). Same in 5d. These are very important information to add in each experiment.

Fig 5e,f: WT and 8A show different levels of expression both in the input and in the IP (e), which may contribute to the reduction of its interaction with the replisome components (IP in e, SIRF in f).

Is USP37-replisome interaction increased upon CPT (as shown by SIRF)? Does the ubiquitin binding ability of USP37 (via the 3XUIMs) play a role?

Fig 5g,h: Since the 8A mutant appears less expressed (Fig. 5e), the reduced complementation could be due to this lower expression.

Line 549-550: 'Note that CTRL+EV and USP37 KO+ mCherry-USP37WT samples are the same as in Fig. 1h and i'.

Unfortunately, neither in Fig. 1h,i nor Fig. 5g,h the USP37 protein levels are shown; they should be provided for each experiment.

Line 172: Typo, it should be MCM7 and MCM4 (and not MCM6).

There are some inconsistencies in the results among different figures. For example, in Fig 4a lane 8 and Fig 4c lane 1, the samples should be treated in the same way (p97i) but the MCM7 blot appears very different in term of proportion of unmodified versus ubiquitinated MCM7.

Also, in Fig. 4a, the MCM6 immunoblot appears very similar in lanes 8,10 (p97i) and 1, 2, which is rather unexpected.

Related to this, at line 243 it stated that 'no ubiquitylation of MCM6 was observed in USP37-depleted extracts', but there is also no ubiquitination upon p97i only, which on the other hand should be detectable as there is no cullin inhibitor.

Reviewer #5

(Remarks to the Author)

Review manuscript NCOMMS-24-55130-T « USP37 prevents premature disassembly of stressed replisomes by TRAIIP.

In their manuscript, the authors identified that the deubiquitylase USP37 counteracts the E3 ubiquitin ligase TRAIIP to avoid aberrant disassembly of the replication forks during interphase using a combination of genetic assays in cell culture cells and biochemical assays in the *Xenopus* in vitro system when forks were blocked with topoisomerase inhibitors. They also used AlphaFoldMultimer to predict the interaction of USP37 with the fork complex CMG component Cdc45 and tested this predicted interaction by mutagenesis of Cdc45-interacting residues in USP36 in the *Xenopus* system and human cells. Their surprising finding that TRAIIP promotes unloading in USP37 depleted *Xenopus* egg extracts when forks are stalled more than 1 kb apart, led to an attractive model on the regulation of CMG unloading and replication termination.

The observations reported in the paper are novel and potentially significant, however, several key experiments require additional statistical tests and normalised western blot quantifications including statistical tests should be done to strengthen their claims and meet the standards of the journal. Authors should also show representative images with foci of key immunofluorescence experiments they quantified in their figures to get an impression of the quality of their labelling and visualise the effects quantified.

Major points:

1. In Fig. 1b-e, the authors show that different USP37 knock-out clones (KO10, KO19) have a lower mean survival curve than the WT in the presence of different drugs from three independent experiments. The means of the two different clones are on separate curves but the errors of mean of the two different USP37 clones overlap. Are the means of the USP37 clones significantly different and if so, why? The authors should provide statistical tests with p-values to formally demonstrate the significant differences in all of these types of figures.

The same remark applies to Fig 1 h-j, Fig.2 c-e and Fig. 5 g-h.

2. The authors should show representative photos for Fig 1f, g, Fig 2 f-g, 3d and 5f to get an impression of the quality of their RPA and GammaH2AX labelling and the described effects.

3. Authors analysed the effects of camptothecin, a topo 1 inhibitor, and ICRF193, a topo II inhibitor, and aphidicolin in cultured cells on cell survival and DNA damage. They then analysed only the effect of ICRF in the *Xenopus* in vitro system. Does USP37 depletion also induce premature CMG unloading in the presence of camptothecin?

4. The authors show data western blots against several components of the CMG in the *Xenopus* in vitro system in large panels Fig 3 a-c, 4 a-e and Fig 5c,d and in extended data. For the reader, it would be easier to follow the results when quantification of the western blot bands, normalised to the loading control histone H3, of the key lanes (as indicated in the text) is provided. This could be done for one of the MCMs or Cdc45. They state in the text that at least 2 replicates have been performed for each experiment, but other western blots of replicates are not provided in the extended data. Authors should perform statistical tests with a p-value to demonstrate significant differences between quantifications of western blot bands in their different replicates. If incubation times in min differ too much between replicates because of differences in extract quality, authors could classify incubation times.

5. The authors analysed the effects after USP37 depletion on CMG unloading in *Xenopus* in vitro system using plasmid DNA as a substrate to replicate in NPE extracts. The obvious advantage of this soluble system becomes clear when plasmid DNA with inserted arrays of varying sizes are used. However, DNA replication dynamics of small plasmids and chromosomal DNA are likely to be different concerning the number of termination events, topology constraints, and fork speed as well as chromatin assembly. Can the authors reproduce some of their key findings of CMG unloading after

chromosomal DNA replication using *Xenopus* sperm nuclei as substrate in low-speed egg extracts depleted of USP37?

Minor points:

-In Fig 1 b-e, different USP37 clone numbers are indicated than in Fig 1. f-g, analysing the cell survival for one set of clones and the RPA or gamma-H2AX foci for another set of clones. Are the behaviours of different clones varying depending on the assay or why did the authors use different clones in these experiments?

-In lane 97, the authors should be more careful when stating « treatment with aphidicolin, at doses that does not induce damage in Ctrl cells... ». Since they cannot exclude that there is damage after even low doses of aphidicolin, which they cannot detect in their assays, they should change the phrase to « at doses that does not induce any detectable damage ».

Version 1:

Reviewer comments:

Reviewer #1

(Remarks to the Author)

The concerns are well addressed in the updated manuscript. The reviewer agrees to publish it as it is.

Reviewer #2

(Remarks to the Author)

Reviewer #3

(Remarks to the Author)

I found the article and timely and important. The authors have addressed all my concerns and the manuscript is now suitable for publication.

Reviewer #4

(Remarks to the Author)

We acknowledge that the manuscript significantly improved after the revisions implemented by the authors, as the necessary controls and additional validations are now included.

Notwithstanding, there are small corrections the authors should make before publication.

Some conclusions sound too strong based on the data presented.

Line 172: 'Premature CMG unloading was suppressed by supplementing USP37-depleted extracts with....'. The use of 'reduced' rather than 'suppressed' is more correct based on the data.

Line 218: 'reducing CMG unloading' should be used instead of 'preventing CMG unloading'.

The labeling in Fig. 2d,e is wrong (circles versus triangle).

The main issue that remains concerns experiments using the C350A mutant. As pointed out previously, the protein levels of the wild type and C350A mutant of USP37 is very different, which makes it hard to draw conclusions on the relevance of catalytic activity on cell survival (Fig. 1i-k). The explanation provided by the authors 'We note that catalytically inactive USP37C350 always shows a lower band and an upper band that we believe is an auto-ubiquitylated form of USP37' is not convincing. It is possible that the higher bands correspond to the ubiquitinated forms of USP37, but usually the ubiquitinated forms are much less prominent than the unmodified, as also shown for USP37 (for example see Tanno et al, 2014, Fig. 3c; doi.org/10.1074/jbc.M113.528372). Moreover, the increased ubiquitination of USP37-C350A compared to wild type, likely due to the lack of auto-deubiquitination activity, makes USP37 more prone to degradation induced by ubiquitin ligases such as SCF and APC complex (DOI 10.1074/jbc.M112.390328; DOI 10.1016/j.molcel.2011.03.027), and it is indeed what the authors showed (Fig. R3). USP37-C350A looks more ubiquitinated and much more degraded. It therefore remains unclear how much USP37-C350A protein is present in these cells, rendering it somewhat difficult to reach strong conclusions on this aspect.

Reviewer #5

(Remarks to the Author)

The authors revised most points that I suggested correctly. I have no objection to publication.

REVIEWER COMMENTS

We thank the reviewers for their detailed and constructive comments. Although we addressed most of the comments and added new data, some comments would have required extensive experimentation that would not significantly alter the conceptual message of the paper. In addition, complementary manuscripts by Labib and Emanuele groups have been recently resubmitted for publication. Given these considerations, we elected not to perform a small number of the requested experiments. We hope the reviewers will nevertheless find the paper much improved and that it advances our understanding of how aberrant TRAIP function at stalled replisomes is restricted by USP37. The reviews are reproduced in their entirety, and our responses are added in blue font.

Reviewer #1 (Remarks to the Author):

Kochenova et al. found through in vivo and in vitro studies that the deubiquitinase USP37 can protect the CMG complex, preventing its premature dissociation from replication forks under replication stress, thereby maintaining genome stability. The authors performed a CRISPR screen to identify which protein deficiencies cause cellular sensitivity to CPT and identified the deubiquitinase USP37. This result was validated using USP37 knockout monoclonal cells. To further understand why the loss of USP37 sensitizes cells to CPT, the authors conducted another CRISPR screen in USP37 knockout cells and discovered that the loss of the TRAIP E3 ligase rescues the sensitivity of USP37-deficient cells to CPT, indicating a functional relationship between USP37 and TRAIP. Previous research in the Walter lab showed that, on interstrand crosslinks, after CMG convergence, TRAIP promotes CMG dissociation by ubiquitinating CMG. Building on these findings, the authors investigated whether USP37 stabilizes the CMG under CPT-induced replication stress by counteracting TRAIP and deubiquitinating MCM. They tested this hypothesis using the *Xenopus* egg extract system. The authors further explored the conditions under which TRAIP ubiquitinates CMG during replication stress and found that approximately 1 kb of distance between two CMGs could trigger CMG ubiquitination, and this triggering mechanism may involve not only the distance but also the chromatin structure between the two CMGs. Finally, the authors used AlphaFold to predict the binding mode and binding sites of USP37 to CMG and performed mutational analysis on these sites. They found that the PH domain of USP37 directly interacts with CDC45. Furthermore, they validated the interaction and function using the *Xenopus* egg extract system. Overall, this study is highly innovative, logically rigorous, well-supported by evidence, and meticulously written.

Major concerns:

1. The authors emphasize that the loss of USP37 causes sensitivity to CPT due to premature CMG dissociation. Please use DNA fiber single-molecule experiments to observe whether replication forks can restart after the removal of CPT and cisplatin treatment.

We now performed DNA fiber analysis following camptothecin washout (Extended Data Fig. 7b) and showed that cells lacking USP37 fail to properly restart stalled DNA replication as monitored by the percentage of stalled forks. The ability of forks to restart

was rescued by loss of TRAIIP. Since we have not investigated the role of USP37 and TRAIIP in dealing with replication damage caused by cisplatin treatment anywhere in the current manuscript, we did not feel that carrying out DNA fibre analysis with cisplatin was appropriate.

2. In Figure 3d, the PLA is for CDC45 and EdU. However, since USP37 knockout cells have replication issues, the EdU incorporation might change. The PLA signal should be normalized to the EdU signal. The same applies to Figure 5f.

We now quantify and show in new Extended Figure 7 and Extended Figure 12a, respectively, the EdU incorporation from samples in Figure 3d and 5f and see no statistically significant difference between any of the conditions tested.

Minor points:

1. The illustrations in Figures 2D and 2E contain errors.

We are sorry but we can't see what the errors this reviewer refers to are. If these are highlighted to us, we will of course rectify any errors.

2. The reference in line 245 needs correction.

We thank the reviewer for pointing out this oversight. We have now corrected it.

Reviewer #2 (Remarks to the Author):

Reviewer #3 (Remarks to the Author):

The scientific work presented here describes the role of USP37 in promoting the deubiquitylation of CMG complexes stalled at replication blocks. This mechanism is crucial to allow the correct execution of DNA replication and prevents genome instability. The manuscript is excellent and uses elegant genetic experiments together with complex in vitro reconstitution assays to derive a clear picture on the role of USP37 and TRAIIP in controlling CMG unloading. The experiments are carefully executed and well-controlled, the literature is discussed in detail and the final model is presented in a crystal-clear manner.

I have minor comments which might help the flow of the manuscript for the non-expert readers.

Line 36 unloading by CRL2Lrr1 (Ref.10). Line 152 Data Fig. 5a; Ref.6).

Authors often use Ref. to indicate References. Is there a particular reason to use this type of format in the above indicated lines?

If a reference is to appear next to a number in the text, for example, following an biological acronym, we use “ref.#” to distinguish the preceding word and the reference, according to Nature style. We will of course, adhere to journal editorial guidelines during the publication process.

[REDACTED]

Figure R1. Gene effect (Chronos) of USP37 loss in various cell lines. A negative score indicates that gene loss negatively affects cellular fitness. Adapted from <https://depmap.org>

USP37 is a common essential gene according to DEPMAP, although this doesn't subtract from the validity of the reported observations, I believe it is important to clarify whether these cells have completely lost USP37 WT. Alternatively, are experiments conducted in cells shortly after sgRNA mediated gene depletion? Can the authors clarify? Phenotypes are validated using rescue/put back of USP37 and a catalytically inactive mutant, thus, there is no question that they depend on USP37 but it can be useful to clarify the genetic identity of the clones.

We thank the Reviewer for pointing this out. While USP37 is a common essential gene, it is in fact not essential in U2OS cells [Gene effect (Chronos): -0.306] (Figure R1). Indeed, we managed to obtain knockout clones with no detectable residual USP37 band. In RPE-1

cells, USP37 seems to be essential [Gene effect (Chronos): -1.24], but if we knocked out TP53, we could recover viable clones, suggesting that cell death was caused by genome instability.

We now changed the text highlighting this point (see lines 96-98).

In figure 2E the authors show that Aphidicolin sensitivity is not rescued by double depletion of TRAIP and USP37. The authors appropriately comment on this in the discussion postulating the presence of a second ligase. Now, in the current position this observation seems to distract rather than enrich the flow. Maybe a rationale for using Aphidicolin in these settings could be briefly mentioned in lines 130-135 (and discussed later).

We have added the phrase in bold: “In contrast, TRAIP loss did not restore USP37 knockout cells' tolerance of aphidicolin (Fig. 2e and Extended Data Fig. 4c), **suggesting a different cause of toxicity when USP37 is lost in this setting.**”

The results of the two genetic screens (Fig.1A and Fig.2A) conducted could be presented in excel tables to help the reader follow up on parallel observations and increase the impact of the work.

The CRISPR screen datasets have been deposited in Dryad, with a link provided in the Data Availability section of the Methods.

The model presented in Figure 6 is crystal clear but doesn't encapsulate the important message from the manuscript about the ubiquitylation of CMG by TRAIP in Cis vs Trans.

If the authors refer back to the model in Extended Figure 1A in the Figure legend of Figure 6, can this facilitate the presentation of this important finding ?

We thank the reviewer for this suggestion and now refer to the Extended Data Fig. 1a in the Discussion, and in the Fig. 6 legend.

Can an explanation of the differential ubiquitination be USP37 unloading from CDC45 based on DNA conformation or fork convergence? This could be one of the reasons why USP37 is not often detected in iPOND and similar techniques?

We assume the reviewer is referring to the fact that USP37 loss is not required to allow TRAIP activity during ICL repair (please compare the MCM7 ubiquitylation, marked with a red line, in Figure R2b). Therefore, we speculate that USP37 does not dissociate from fully converged CMGs. However, USP37 does not seem to significantly affect the kinetics of CMG unloading or ICL repair (as determined by the resolution of "Figure 8" structures in the replication in Figure R2c). We interpret this as follows: when CMGs are fully converged on an ICL, TRAIP is positioned such that it can ubiquitylate CMGs efficiently, and that the rate of ubiquitylation is faster than de-ubiquitylation by USP37. We prefer not to include these results in the manuscript because they have to be repeated and may form part of another manuscript.

Reviewer #4 (Remarks to the Author):

CMG (Cdc45-Mcm2-7-GINS) helicase disassembly is a crucial event in DNA replication, playing a key role in maintaining genome integrity by preventing re-replication. A key process regulating CMG disassembly is ubiquitination, largely occurring on the MCM7 subunit of the MCM complex and leading to unfolding by the p97/VCP ATPase. There are two major E3 ubiquitin ligases regulating CMG-MCM7 ubiquitination and disassembly, CRL2LRR1 and TRAIP. During replication termination, CMGs unloading from chromatin

Figure R2. The role of USP37 in ICL repair. **a**, A schematic showing converged CMGs at cisplatin ICL (blue line) being ubiquitylated by TRAIP. **b**, Plasmid DNA containing cisplatin ICL was incubated in the indicated egg extracts in the presence or absence of 200 μ M p97-i. At specified times, chromatin was recovered and immunoblotted for the indicated proteins. **c**, Plasmid DNA was replicated in the presence or absence of USP37 in extracts containing [α - 32 P]dATP. Replication intermediates were separated on a native agarose gel and visualized by autoradiography

is promoted by CRL2LRR1, which also removes CMGs at specific types of damage. On the other hand, TRAIP associates with the replisomes and ubiquitinates CMGs at inter-strand cross-links (ICLs), promoting unloading of CMGs by p97. During S phase, TRAIP apparently ubiquitinates CMGs, and also DNA protein cross-links (DPCs) in trans, and is unable to ubiquitinate the replisome in cis, which is crucial to prevent premature unloading

of CMGs. In mitosis, TRAIP promotes in cis ubiquitination of CMGs and disassembly of the replisome to allow chromosome segregation. Keeping the activity of TRAIP (and CRL2LRR1) under tight control is crucial to avoid deregulated ubiquitination and disassembly of the CMG complex in interphase.

Figure R3. Western blot validation of mCherry-USP37^{WT}, mCherry-USP37^{C350A} (catalytically inactive), or mCherry-USP37^{8A} expression in USP37 knockout cells. Quantification of the mCherry-USP37 band relative to the WT is shown in red.

The present manuscript aims to identify new regulators of DNA topological stress, by a genome wide CRISPR screen in U2OS upon TOP1 inhibitor (CPT). Among the factors whose inactivation causes CPT hypersensitivity, the authors focused their studies on the DUB USP37 by combining cell sensitivity assays in human cells and mechanistic studies in *Xenopus* egg extracts. The study is interesting and timely, addresses important mechanisms at the basis of the regulation of the CMG complex disassembly and identified a new factor regulating this process.

Overall, the work is well organized, and the results are important. However, in some cases the data obtained were not strong enough to support the conclusions drawn, and there are few important points that need to be addressed to make it competitive for publication in *Nature Communications*. These key points are discussed below.

Major comments

General

In the manuscript, *Xenopus* egg extracts have been widely used to measure CMG unloading (Fig. 3-5), and for non-experts of this system it can be difficult to appreciate how it works. It would be helpful to provide more explanation in the main text, supported perhaps by a schematic drawing, to facilitate understanding of the results.

We thank the reviewer for this suggestion, and in Extended Data Fig. 1a, we have now added a schematic explaining the experimental flow using egg extracts.

Furthermore, as several cell lines are used in the study, it would be beneficial to the reader if the cell line used was indicated above a figure.

We now specify more clearly the cell line used for each experiment.

Another important point, for most of the survival assays, the protein levels of USP37 variants (WT and mutants) are not shown, making it difficult to draw conclusions on the

effect of complementation and the relevance of each domain (i.e., RING finger and PH) in USP37 activity.

In the original manuscript, we showed expression of USP37^{WT}, USP37^{C350}, and USP37^{8A} (PH mutant) in RPE-1 TP53KO USP37 KO10 in Extended Data. Fig. 2i. We now also show the expression of these constructs in the USP37 KO19 clone. We note that catalytically inactive USP37^{C350} always shows a lower band and an upper band that we believe is an auto-ubiquitylated form of USP37. Quantification (shown in red in Figure

R3) of the single USP37^{WT} and USP37^{8A} band and the double USP37^{C350} band shows similar levels of expression.

Also, it is known that USP37 contains 3 ubiquitin interacting motifs (3XUIMs) that mediate the binding to ubiquitin and also regulate its catalytic activity (Manczyk et al, Sci Rep 2019). What is the role of these 3XUIMs in the association with ubiquitinated components of the replisome is missing. For the sake of clarity, the possible contribution of the UIM domains needs to be clarified and tested.

In the cell survival assay, we observed that the 3xUIM mutant fully complemented USP37 KO (Figure R4), which is consistent with previous findings that this mutant induces “a decrease in the steady state pool of ubiquitinated proteins in the nucleoplasm, suggesting that USP37 exhibits DUB activity in the nucleus” (see Fig. 2, panel A, B, and F, FK2 staining for ubiquitin in PMID: 24324262). We agree with the Tanno et al. interpretation that cellular overexpression of the 3xUIM mutant likely masks the reduced catalytic activity observed in in vitro experiments. Regarding the interaction with the replisome, we show that USP37 interacts with the replisome via the PH domain even before MCM7 is ubiquitylated by TRAIP, suggesting that UIM domains are likely to be dispensable in the absence of the available ubiquitin chain.

Specific points

Fig. 1h-j: The authors measured the survival of USP37 KO cells and the effect of complementation with USP37 WT and catalytic mutant C350, showing that the C350 mutant does not restore viability of cells upon different types of stress. This experiment is problematic, because the expression of the C350A is very poor compared to WT (as shown in Ext. Fig. 2i) and therefore it is not possible to draw any conclusion from this experiment. To unequivocally prove the need of USP37 enzymatic activity in the survival of cells exposed to genotoxic stress, the experiment should be repeated using cells expressing equal (or at least similar) levels of USP37 WT and C350.

Lines 107-108: ‘These data indicated that the catalytic activity of USP37 protects cells from the deleterious effects of replication fork stalling’. Based on the levels of C350 mutant complementation, this cannot be stated.

Figure R4. Clonogenic survival assays of control (CTRL) cells or USP37 KO19 cells complemented with vectors expressing mCherry (E.V.), mCherry-USP37^{WT}, mCherry-USP37^{UIM} upon treatment with camptothecin (CPT). n=3 independent experiments. Bars represent means ± SEM.

USP37^{C350} always shows a lower band and an upper band that we believe is a ubiquitylated form of USP37. Quantification (shown in red in Figure R3) of the single USP37^{WT} and the double USP37^{C350} band shows similar levels of expression.

Lines 99-100: 'these data support a role for USP37 in promoting genome integrity upon treatment with CPT and aphidicolin'. These data do not indicate a role of USP37 in promoting genome integrity but rather in cell survival.

In lines 99-100, we referred to Fig. 1f-g (current Fig. 1g-h), which quantifies RPA and γ H2AX foci. Since RPA and γ H2AX foci are markers of ssDNA and DSB, respectively, we believe we can conclude that these data support a role for USP37 in promoting genome integrity upon treatment with CPT and aphidicolin rather than a role in cell survival.

Fig. 3a, Lines 162-163: 'While USP37 depletion did not influence CMG unloading during unperturbed replication....'. Difficult to claim this, since the drop in CMG unloading was very drastic (from 10' to 45'). It would be of help to test this using intermediate timepoints.

We have now repeated this experiment with early time points for untreated conditions (see Fig. 3a and Extended Data Fig. 6b). However, despite have a reproducible effect on CRL2^{Lrr1}-dependent ubiquitylation, USP37 depletion doesn't seem to have a reproducible effect on the kinetics of CMG unloading in the absence of ICRF-193 (compare Fig. 3a; Extended Data Fig. 5d, lanes 7-12; Extended Data Fig. 6b, Figure R5). Therefore, we concluded that USP37 depletion has minimal impact on CMG unloading during normal termination. We speculate that the mechanism of CRL2^{Lrr1} binding to the terminated replisome may already facilitate optimal levels of ubiquitylation and unloading. As a result, the absence of USP37 may not provide an additional "boost" in chain length that would further enhance this process.

Also, the difference between lane 9 and 12 is not so convincing (especially for MCM6). The difference is clear for MCM7, CDC45, TRAIPI, and the unmodified form of MCM6, so we believe our conclusion is robust. We occasionally see the slower migrating MCM6 species, but this modification appears to be TRAIPI-independent (see Extended Data Fig. 6f. Lanes 9-16). For the sake of full documentation of our results, we do not want to remove that species.

There is a difference between MCM6 and MCM7 ubiquitylation after 110 minutes (lane 12). How would the authors explain this?

Figure R5. CMG unloading during normal termination in the absence or presence of USP37. Top, Western blot analysis of mock and USP37 depletions. Bottom, Plasmid DNA was incubated in the indicated egg extracts. At the specified times, chromatin was recovered and blotted for the indicated proteins.

We have previously shown that TRAIPI ubiquitylates multiple CMG subunits and to a different extent (Wu et al., 2019 Nature). The extent of ubiquitylation of different subunits could be explained by the difference in the distance between TRAIPI's RING domain and the availability of surface-exposed Lysine residues on different subunits: the closer it is positioned relative to TRAIPI - the more efficiently it is ubiquitylated. It could be also related to the conformation two CMGs adopt during fork convergence or the fact that TRAIPI is a dimer (PMID:26093298; predictomes.org).

Fig. 3b: The effect of p97i on the unloading of the fork seems quite small, especially since the blot without the p97i is on another gel, making it difficult to take strong conclusions.

The role of p97i in CMG unloading in egg extracts is well established (Moreno et al., 2014 Science; Dewar et al., 2017 G&D; Wu et al., 2019 Nature), and is visible here as well (compare TRAIPI and CDC45 levels in lanes 4 and 12). The reason the effect is less pronounced in this experiment is due to the presence of ICRF-193, which on its own greatly slows CMG unloading (Fig. 3a, mock conditions). The major point of Fig. 3b is to use p97-i to trap ubiquitylated CMG on chromatin and to compare the extent of ubiquitylation in the presence or absence of WT and C347S USP37 (lanes 9-12 within the same gel) as detected by the amount of unmodified MCM7 and MCM4.

Fig. 5g,h: The authors show that USP37 binds CDC45 and that TRAIPI loss is beneficial to USP37 knockout cells. As the authors state that their data suggest that USP37's interaction with CDC45 counteracts TRAIPI mediated CMG ubiquitylation, it would be nice if the authors include sgTRAIPI conditions.

The sgTRAIPI condition is included in Figures 1c and 1d and demonstrates that TRAIPI loss also confers camptothecin sensitivity. We speculate that in TRAIPI's absence, some aspect of the response to CPT is defective, and that this response is rescued by USP37 knockout because now, another E3 ligase becomes active enough to substitute for TRAIPI. As suggested by the reviewer, we now transduced USP37 KO cells with either a control sgRNA or an sgRNA targeting TRAIPI, selected for stably transduced cells, and monitored editing efficiency using TIDE (Extended Data Fig. 12e). As expected, expression of USP37^{WT} in USP37 KO cells restored viability upon camptothecin treatment, whereas neither the empty vector nor the USP37^{8A} mutant was effective (Fig. 5i). Consistent with our previous findings, TRAIPI loss was toxic in WT cells (USP37 KO cells complemented with USP37^{WT}). However, TRAIPI loss conferred a survival advantage to USP37 KO cells expressing the empty vector or the USP37^{8A} mutant, which is unable to bind CDC45 and the replisome. These data further suggest that when USP37 is unable to locate to the replisome, TRAIPI loss is beneficial as the ubiquitylation events TRAIPI mediates can't be countered.

Figure R6. Quantification of S-phase (EdU positive) U2OS WT or USP37 KO cells

Minor comments

For the results in Fig.1f,g, Fig.2f,g, Fig. 3d, it would help to see the variability in the assay by showing the median of each replicate in the figure.

We now show the median of each replicate, as requested by the reviewer.

Do the USP37-KO cells have a normal cell cycle distribution? Expect lower EdU incorporation for USP37-deficient cells, do the authors also see this in the data from Fig. 1f?

We now quantified the percentage of S phase (EdU positive) cells and observed a minor, not statistically significant, difference between WT and USP37 KO U2OS cells (Figure R6). Similarly,

no significant difference in EdU incorporation was observed between CTRL and USP37 KO RPE-1 cells (see untreated conditions from current Extended Data Fig. 7a).

Figure R7. Quantification of CDC45 levels in the main Fig. 3b.

Fig. 1b-d: Why do the RPE1 clones (KO10 and KO19) behave so differently, since the KO looks efficient in both? Does this indicate a clonal variability? Along the same line, why do the U2OS clones (KO6 and KO17; Fig. 1f,g) behave differently in untreated conditions?

Although the two RPE-1 USP37 KO clones show a difference upon treatment with some DNA-damaging agents, this is not statistically significant. In untreated conditions, U2OS USP37 KO clone 6 shows slightly higher levels of RPA compared to clone 17. However, the same difference is not observed for γH2AX foci. Therefore, we believe this is a clonal effect that does not impact our findings.

Fig. 2f,g: Surprisingly the siTRAIP treated control cells show less γH2AX foci and RPA foci upon CPT treatment, however loss of TRAIP is toxic upon increasing doses of CPT in Fig. 2c. How do the authors explain this?

We also found, surprising, that TRAIP loss leads to reduced RPA and γH2AX foci, while being toxic upon camptothecin treatment. Hoffmann, 2016 (DOI: [10.1083/jcb.201506071](https://doi.org/10.1083/jcb.201506071)) observe a similar reduction in CPT-induced ssDNA formation and RPA loading upon TRAIP depletion. This might reflect failure of a repair pathway that involves a transient ssDNA intermediate. Alternatively, or in addition, TRAIP knockdown might shift the balance towards toxic repair pathways, like toxic NHEJ, which we have shown in the past to cause cell death upon treatment with CPT but that avoid ssDNA generation (Balmus, 2019).

Fig 3b, line 174: 'Furthermore, this effect was reversed by USP37 WT but not USP37-C347S (Fig. 3b, lanes 14 and 16)'. The effect of the WT complementation is not very strong, the suggestion is to tone it down.

We respectfully disagree with the reviewer's assessment that the WT complementation effect is not very strong. Instead, our data suggest that the WT protein effectively rescues

CMG unloading to levels comparable to those seen in the mock extract (Figure R7). To further support this, we have also included a second replicate in Extended Data Fig. 6d. Therefore, we would prefer to maintain our current wording.

Fig 4d, e: How are the meDPCs induced here?

Methylated M.HpaII was covalently linked to plasmid DNA by including 5-fluoro-2'-deoxycytosine in its recognition site, CCGG, which is a methylation suicide substrate (Duxin et al., 2014 Cell).

How can TRAIP promote ubiquitination at distant CMGs?
Not sufficient data in support of this point.

We suspect it has to do with replisome dimerization, as recently proposed (PMID: 38484065), but despite extensive efforts, we have not been able to show replisome-replisome associations in extracts. Figuring this out will take time, and we do not want to delay publication of the other conclusions, which we believe will be of great interest to the community.

Page 10 lines 263-264: How do you know that in your experiments, MCM7 ubiquitination in mitosis is completely dependent on TRAIP? It would be advisable to test depletion of TRAIP in experimental setting as in Fig 4d-e and prove this point.

In previous published and recently submitted work, we find that CMG unloading in mitotic extracts is always TRAIP-dependent (Deng et al., 2019 Mol Cell; <https://www.biorxiv.org/content/10.1101/2024.11.30.626186v1>). To address the reviewer's point, we will explicitly state the inference of TRAIP-dependence in the current experiments.

USP37-8A mutant has reduced enzymatic activity compared to WT (Ext fig 7j), which should be mentioned in the text.

We now mention this fact in the text and refer to a relevant quantification of the effect in the new Extended Data Fig. 10l.

Fig. 5a: The authors need to show USP37 levels in the whole cell lysate upon CDKi, as this could affect the expression of USP37 and could therefore also explain the lack of binding to the DNA.

As requested, we have now included the data showing that USP37 is not degraded in mitotic extracts (see Extended Data Fig. 9j).

Fig. 5c: Introducing 8 mutations might have quite a dramatic impact on the structure of the PH-domain, with unpredictable effect on the folding of the whole protein. It would be preferable, if possible, to generate a deficient mutant by introducing fewer amino acid alterations.

Figure R8. Extracts of HEK293T cells expressing mCherry (empty vector, EV) or mCherry-USP37^{WT} or mCherry-USP37^{8A} were subjected to mCherry immunoprecipitation (IP) followed by western blotting for the indicated proteins. This experiment was repeated three times with similar results. Quantification of the mCherry-usp37 band is shown in red.

We now included USP37 variants with fewer amino acid changes (4A-L and 4A-R), which show similar results to the 8A mutant.

Fig 5c, lane 8: Again, the levels of expression on WT and 8A are not shown. In the input samples, only 1 lane is shown (and not indicated whether it corresponds to the WT or the 8A sample). Same in 5d. These are very important information to add in each experiment.

We agree this is important: the expression levels for WT and 8A proteins were shown originally in Extended Figure 7k and l, and now in Extended data Fig. 11a and c.

Fig 5e,f: WT and 8A show different levels of expression both in the input and in the IP (e), which may contribute to the reduction of its interaction with the replisome components (IP in e, SIRF in f).

We now quantified the expression of USP37^{WT} and USP37^{8A} in the IP experiment (Fig. 5e), as indicated by the numbers in red in Figure R8. We believe such a small difference in expression could not be responsible for an almost complete loss of interaction, as seen for most of the interacting proteins.

Regarding the SIRF experiments (Fig. 5f), these are performed in USP37 KO cells stably expressing USP37^{WT} or USP37^{8A}, whose expression is not changed as shown in Extended Data Fig. 2i.

Is USP37-replisome interaction increased upon CPT (as shown by SIRF)?

Thank you for pointing this out. We now mention this in the manuscript in lines 340-342.

Does the ubiquitin binding ability of USP37 (via the 3XUIMs) play a role?

This is an interesting possibility. However, given our negative data with 3xUIM mutant in cell-based assay (Figure R4) and difficulty with detecting the USP37 binding increase upon ICRF-193 treatment in egg extracts by western blotting (Figure R9), we decided not to pursue this further.

Fig 5g,h: Since the 8A mutant appears less expressed (Fig. 5e), the reduced complementation could be due to this lower expression.

This experiment was done in RPE-1 USP37 KO cells complemented with WT or 8A USP37, which are equally expressed (Figure R3).

Line 549-550: 'Note that CTRL+EV and USP37 KO+ mCherry-USP37WT samples are the same as in Fig. 1h and i'. Unfortunately, neither in Fig. 1h,i nor Fig. 5g,h the USP37 protein levels are shown; they should be provided for each experiment.

ICRF-193 was added into replication reaction at 5 min

Figure R9. *Xenopus* sperm chromatin was replicated in the indicated extracts in the presence of 200 μ M ICRF-193. At the specified times, chromatin was recovered and immunoblotted for the indicated proteins.

The protein levels of the stable cell line used for experiments in Fig. 1h,i and Fig. 5g,h are shown in Extended Data Fig. 2i.

Line 172: Typo, it should be MCM7 and MCM4 (and not MCM6).
Thank you for pointing this out. We have now corrected it.

There are some inconsistencies in the results among different figures. For example, in Fig 4a lane 8 and Fig 4c lane 1, the samples should be treated in the same way (p97i) but the MCM7 blot appears very different in term of proportion of unmodified versus ubiquitinated MCM7.

The difference in the ubiquitylation pattern of MCM7 in Fig. 4a lane 8 and Fig 4c lane 1 is explained by the activity of two different E3 ubiquitin ligases in these assays. In Fig 4a lane 8, two converging CMGs are stalled at DPCs, and because USP37 is not depleted, MCM7 is not efficiently ubiquitylated by TRAIP. Therefore, there is not much ubiquitylation of MCM7. In Fig.4c lane 1, CMGs were not stalled and, thus, completed DNA replication. During replication termination, CRL2^{Lrr1}, is specifically recruited to ubiquitylate MCM7. Unlike ubiquitylation by TRAIP, CRL2^{Lrr1}-dependent ubiquitylation is characterized by the presence of a prominent band corresponding to chains of ~5-7 ubiquitins. Also, in Fig. 4a, CRL2^{Lrr1} cannot ubiquitylate stalled CMGs as its binding site is occluded by the lagging strand (Jenkyn-Bedford et al., 2020 Nature; Low et al., 2020 G&D).

Also, in Fig. 4a, the MCM6 immunoblot appears very similar in lanes 8 ,10 (p97i) and 1, 2, which is rather unexpected.

Lane 10 appears to be slightly underloaded in Fig 4a. We have now added the second replicate of this experiment in Extended Data Fig. 8b, where it is clear that the amount of unmodified MCM6 is less in lane 10 than in lanes 8 and 1-2.

Related to this, at line 243 it stated that 'no ubiquitylation of MCM6 was observed in USP37-depleted extracts', but there is also no ubiquitination upon p97i only, which on the other hand should be detectable as there is no cullin inhibitor.

We do not expect to detect any MCM6 ubiquitylation in Fig 4c with or without Cullin inhibitor, because, unlike TRAIP, CRL2^{Lrr1} (the relevant E3 in this setting) is more specific and ubiquitylates only MCM7 (Maric et al., 2017 Cell Reports; Wu et al., 2019 Nature; Deegan et al., 2020 eLife; Low et al 2020 G&D). Such specificity could be explained by a precise positioning of the RBX1 subunit of CRL2^{Lrr1} across from the preferred ubiquitylation sites on MCM7 (K28, K29) in the CRL2^{Lrr1}-CMG complex (Fig. 3a in Jenkyn-Bedford et al., 2020 Nature).

Reviewer #5 (Remarks to the Author):

Review manuscript NCOMMS-24-55130-T « USP37 prevents premature disassembly of stressed replisomes by TRAIP.

In their manuscript, the authors identified that the deubiquitylase USP37 counteracts the E3 ubiquitin ligase TRAIP to avoid aberrant disassembly of the replication forks during

interphase using a combination of genetic assays in cell culture cells and biochemical assays in the *Xenopus* in vitro system when forks were blocked with topoisomerase inhibitors. They also used AlphaFoldMultimer to predict the interaction of USP37 with the fork complex CMG component Cdc45 and tested this predicted interaction by mutagenesis of Cdc45-interacting residues in USP36 in the *Xenopus* system and human cells. Their surprising finding that TRAIP promotes unloading in USP37 depleted *Xenopus* egg extracts when forks are stalled more than 1 kb apart, led to an attractive model on the regulation of CMG unloading and replication termination.

The observations reported in the paper are novel and potentially significant, however, several key experiments require additional statistical tests and normalised western blot quantifications including statistical tests should be done to strengthen their claims and meet the standards of the journal. Authors should also show representative images with foci of key immunofluorescence experiments they quantified in their figures to get an impression of the quality of their labelling and visualise the effects quantified.

Major points:

1. In Fig. 1b-e, the authors show that different USP37 knock-out clones (KO10, KO19) have a lower mean survival curve than the WT in the presence of different drugs from three independent experiments. The means of the two different clones are on separate curves but the errors of mean of the two different USP37 clones overlap. Are the means of the USP37 clones significantly different and if so, why? The authors should provide statistical tests with p-values to formally demonstrate the significant differences in all of these types of figures.

The same remark applies to Fig 1 h-j, Fig.2 c-e and Fig. 5 g-h.

We conducted statistical tests and now provide p-values, as requested by this reviewer. This analysis shows that the difference observed between the two USP37 KO clones upon treatment with certain DNA-damaging agents is not statistically significant.

2. The authors should show representative photos for Fig 1f, g, Fig 2 f-g, 3d and 5f to get an impression of the quality of their RPA and GammaH2AX labelling and the described effects.

We now provide representative images of RPA and γ H2AX staining in Fig 1f and Fig 2f and PLA foci in Fig. 3d and 5f.

3. Authors analysed the effects of camptothecin, a topo 1 inhibitor, and ICRF193, a topo II inhibitor, and aphidicolin in cultured cells on cell survival and DNA damage. They then analysed only the effect of ICRF in the *Xenopus* in vitro system. Does USP37 depletion also induce premature CMG unloading in the presence of camptothecin?

We indeed tried to use Topo I inhibitor, topotecan (an analog of camptothecin) in egg extracts. However, as reported by others

[REDACTED]

Similarly, in plasmid-pulldown assays, we haven't detected any prolonged CMG stalling in the presence of Topotecan or premature CMG unloading in the absence of Usp37 (see Figure R10b). CMGs were unloaded normally by 45 min and the ubiquitylation pattern of Mcm7 and the lack of ubiquitylation of other CMG subunits (Mcm6 and Sld5) were indicative of CRL2^{Lrr1} activity.

TRAIP seems to require a prolonged stalling of CMGs at either trapped Topo II or DPCs. Therefore, we envision several explanations for why inhibition of Topo I doesn't have much effect on fork stalling in egg extracts:

1. Topo I is not a predominant topoisomerase in egg extracts, and, thus, there are not enough trapped Topo I molecules on chromatin that can stall CMGs.
2. It is possible that Topo I adducts are more easily removed/bypassed than Topo II adducts in egg extracts.

4. The authors show data western blots against several components of the CMG in the *Xenopus* in vitro system in large panels Fig 3 a-c, 4 a-e and Fig 5c,d and in extended data. For the reader, it would be easier to follow the results when quantification of the western blot bands, normalised to the loading control histone H3, of the key lanes (as indicated in the text) is provided. This could be done for one of the MCMs or Cdc45. They state in the text that at least 2 replicates have been performed for each experiment, but other western blots of replicates are not provided in the extended data. Authors should perform statistical tests with a p-value to demonstrate significant differences between quantifications of western blot bands in their different replicates. If incubation times in min differ too much between replicates because of differences in extract quality, authors could classify incubation times.

We have now provided the second replicates for all western blot data in main figures in Extended Data Figures. We have also quantified and performed statistical tests for experiments in Extended Data Fig 10k-l. We would also like to clarify that all experiments in *Xenopus* egg extracts were performed at least three times, with the exception of experiments shown in Extended Data Fig. 5b and 9c, which were performed twice as they are a confirmation of previously published results (Heintzman et al., 2019 Cell Rep; Sparks et al., 2019 Cell; Larsen et al 2019 Mol Cell). We now also added this statement to the manuscript.

[REDACTED]

Figure R10. Topoisomerase I inhibitor, topotecan, does not stall replication forks in egg extracts. a, Plasmid DNA was replicated in the presence or absence USP37 in extracts containing [α - 32 P]dATP. Where indicated, extracts were supplemented with Topotecan or ICRF-193 to a final concentration of 200 μ M. Replication intermediates were separated on a native agarose gel and visualized by autoradiography. b, Plasmid DNA was incubated in the indicated egg extracts in the presence or absence of 200 μ M Topotecan and p97-i. At specified times, chromatin was recovered and immunoblotted for the indicated proteins.

5. The authors analysed the effects after USP37 depletion on CMG unloading in *Xenopus* in vitro system using plasmid DNA as a substrate to replicate in NPE extracts. The obvious advantage of this soluble system becomes clear when plasmid DNA with inserted arrays of varying sizes are used. However, DNA replication dynamics of small plasmids and chromosomal DNA are likely to be different concerning the number of termination events, topology constraints, and fork speed as well as chromatin assembly. Can the authors reproduce some of their key findings of CMG unloading after chromosomal DNA replication using *Xenopus* sperm nuclei as substrate in low-speed egg extracts depleted of USP37?

As requested, we examined the effect of USP37 depletion in the context of sperm chromatin replicating in low-speed egg extracts. However, in combination with the immunodepletion procedure, ICRF-193 negatively affected nuclear growth and efficiency of DNA replication, making it impossible to examine the effect USP37 on CMG in this setting. Instead, we used HSS-NPE mix to replicate *Xenopus* sperm chromatin and showed that USP37 also restrains TRAPPC2 during replication of chromosomal DNA in the presence of ICRF-193 (see Extended Data Fig. 6h-k).

Minor points:

-In Fig 1 b-e, different USP37 clone numbers are indicated than in Fig 1. f-g, analysing the cell survival for one set of clones and the RPA or gamma-H2AX foci for another set of clones. Are the behaviours of different clones varying depending on the assay or why did the authors use different clones in these experiments?

These are different clone numbers because they come from different cell lines, U2OS (KO6 and 17) and RPE-1 (KO10 and 19). In general, we find that U2OS are the best cell line for visualising and quantifying DNA damage markers, such as RPA and γ H2AX foci in this case. We provide viability data in both RPE-1 and U2OS cells. We now specify more clearly in the figures which cell line is used.

-In lane 97, the authors should be more careful when stating « treatment with aphidicolin, at doses that does not induce damage in Ctrl cells... ». Since they cannot exclude that there is damage after even low doses of aphidicolin, which they cannot detect in their assays, they should change the phrase to « at doses that does not induce any detectable damage ».

We now changed the sentence according to this reviewer's suggestion.

REVIEWERS' COMMENTS

The reviews are reproduced in their entirety, and our responses are added in blue font.

Reviewer #1 (Remarks to the Author):

The concerns are well addressed in the updated manuscript. The reviewer agrees to publish it as it is.

Reviewer #2 (Remarks to the Author):

Reviewer #3 (Remarks to the Author):

I found the article and timely and important. The authors have addressed all my concerns and the manuscript is now suitable for publication.

Reviewer #4 (Remarks to the Author):

We acknowledge that the manuscript significantly improved after the revisions implemented by the authors, as the necessary controls and additional validations are now included.

Notwithstanding, there are small corrections the authors should make before publication.

Some conclusions sound too strong based on the data presented.

Line 172: 'Premature CMG unloading was suppressed by supplementing USP37-depleted extracts with....'. The use of 'reduced' rather than 'suppressed' is more correct based on the data.

We have changed the text accordingly.

Line 218: 'reducing CMG unloading' should be used instead of 'preventing CMG unloading'.

We have changed the text accordingly.

The labeling in Fig. 2d,e is wrong (circles versus triangle).

Thank you for pointing out this oversight that we have now corrected.

The main issue that remains concerns experiments using the C350A mutant. As pointed out previously, the protein levels of the wild type and C350A mutant of USP37 is very different, which makes it hard to draw conclusions on the relevance of catalytic activity on cell survival (Fig. 1i-k). The explanation provided by the authors 'We note that catalytically inactive USP37C350 always shows a lower band and an upper band that we believe is an auto-ubiquitylated form of USP37' is not convincing. It is possible that the higher bands correspond to the ubiquitinated forms of USP37, but usually the ubiquitinated forms are much less prominent than the unmodified, as also shown for USP37 (for example see Tanno et al, 2014, Fig.

3c; [//doi.org/10.1074/jbc.M113.528372](https://doi.org/10.1074/jbc.M113.528372)). Moreover, the increased ubiquitination of USP37-C350A compared to wild type, likely due to the lack of auto-deubiquitination activity, makes USP37 more prone to degradation induced by ubiquitin ligases such as SCF and APC complex (DOI 10.1074/jbc.M112.390328; DOI 10.1016/j.molcel.2011.03.027), and it is indeed what the authors showed (Fig. R3). USP37-C350A looks more ubiquitinated and much more degraded. It therefore remains unclear how much USP37-C350A protein is present in these cells, rendering it somewhat difficult to reach strong conclusions on this aspect.

We have acknowledged this possibility in the manuscript and cited relevant papers.

Reviewer #5 (Remarks to the Author):

The authors revised most points that I suggested correctly. I have no objection to publication.